

# Identifying the connective strength between model parameters and performance criteria

Björn Guse[1,*], Matthias Pfannerstill[1], Abror Gafurov[2], Jens Kiesel[3,1], Christian Lehr[4,5], and
Nicola Fohrer[1]

[1]Christian-Albrechts-University of Kiel, Institute of Natural Resource Conservation, Department of Hydrology and Water Resources Management, Kiel, Germany
[2]GFZ German Research Centre for Geosciences, Potsdam, Germany
[3]Leibniz-Institute of Freshwater Ecology and Inland Fisheries (IGB), Berlin, Germany
[4]Leibniz Centre for Agricultural Landscape Research (ZALF), Institute of Landscape Hydrology, Muencheberg, Germany
[5]University of Potsdam, Institute for Earth and Environmental Sciences, Potsdam, Germany

*Correspondence to:* Björn Guse (*bguse@hydrology.uni-kiel.de)

**Abstract.**

In hydrological models, parameters are used to adapt the model to the conditions of the catchments. Hereby, the parameters need to be identified based on their role in controlling the hydrological behaviour in the model. For parameter identification, multiple and complementary performance criteria are used, which have to capture the different aspects of hydrological response of catchments. A reliable parameter identification depends on how distinctly a model parameter can be assigned to one of the performance criteria.

We introduce an analysis that reveals the connective strength between model parameters and performance criteria. The connective strength assesses the intensity in the interrelationship between model parameters and performance criteria. In our analysis of connective strength, model simulations are carried out based on a Latin Hypercube sampling. Ten performance criteria including the NSE, the KGE and its three components (alpha, beta and r) as well as the RSR for different segments of the flow duration curve (FDC) are calculated.

With a joint analysis of two regression trees (RT), it is derived how a model parameter is connected to the different performance criteria. At first, RTs are constructed using each performance criteria as target variable to detect the most relevant model parameters for each performance criteria. A second RT approach using each parameter as target variable detects which performance criterion is impacted by changes in parameter values. Based on this, appropriate performance criteria are identified for each model parameter.

A high bijective connective strength is calculated for low and mid flow conditions. Moreover, the RT analyses emphasise the benefit of an individual analysis of the three components of the KGE and of the FDC segments. It is emphasised under which conditions these performance criteria provide insights into a precise parameter identification. Separate performance criteria are required to identify dominant parameters on low and mid flow conditions, whilst the number of required performance criteria for high flows increases with the process complexity in the catchment. Overall, the analysis of the connective strength using RTs contribute towards a better handling of parameters and performance criteria in hydrological modelling.





## 1 Introduction

In models, hydrological processes are represented in a simplified way. Fluxes and changes in states are described by mathematical equations. To adapt the model to the hydrological conditions of the study catchments, multiple parameters are included in the model structure. Each of them has a specific role by representing one or multiple processes.

For reliable model simulations, it is required to identify parameter values that lead to a reasonable reproduction of their corresponding hydrological processes (Wagener et al., 2003; Pfannerstill et al., 2014a). Typically, model parameters are estimated using performance criteria which minimise the differences between measured and modelled discharge. In this parameter identification, it is implicitly assumed that the model parameters are precisely identified by the selected set of performance criteria. To investigate the correctness of this assumption, the interrelationship between model parameters and performance

criteria needs to be identified as an initial step towards accurate parameter identification.

In order to capture all aspects of the hydrological system, it is recommended to use multiple and contrasting performance criteria (Gupta et al., 1998; Vrugt et al., 2003; Krause et al., 2005; Gupta et al., 2009; Reusser et al., 2009; Guse et al., 2014). In this context, we use the term 'performance criteria' as an overall term both for statistical performance metrics and signature measures. Typical statistical performance metrics are the Nash-Sutcliffe Efficiency (NSE) (Nash and Sutcliffe, 1970) and as

a further development, the Kling-Gupta-Efficiency (KGE) (Gupta et al., 2009; Kling et al., 2012) which separately considers the three components bias (KGE_beta), variability (KGE_alpha) and correlation (KGE_r) to improve the estimation of the performance error compared to the NSE.

Recent studies recommended to use signature measures, which are directly related to catchment functions with the aim to consider the relevance of a certain process individually (Yilmaz et al., 2008; van Werkhoven et al., 2009; Clark et al.,

2011; Martinez and Gupta, 2011; Pokhrel et al., 2012; McMillan et al., 2014). Signature measures based on flow duration curves (FDC) provide diagnostic information of how a model performs for different discharge magnitudes (Yilmaz et al., 2008; Cheng et al., 2012; Yaeger et al., 2012; Pfannerstill et al., 2014a). Pfannerstill et al. (2014a) showed that a separation of the flow duration curve into five segments improved the model results for different discharge magnitudes and reduced the trade-off between satisfying results both for high and low flows in the same model run. By using different signature measures, the

hydrologic behaviour is represented better in the performance assessment (Martinez and Gupta, 2011; Singh et al., 2011; Euser et al., 2013) and a precise interpretation which processes are accurately reproduced is realised (Gupta et al., 2009).

Each performance criterion emphasises different hydrological conditions with respect to e.g. discharge dynamics, discharge magnitude, water balance, high flows (Madsen, 2000; Boyle et al., 2001; Wagener et al., 2001). By selecting a specific performance criteria, a certain part of the hydrograph is inevitably weighted higher than other parts (Yapo et al., 1998; Madsen et al.,

2002; Vrugt et al., 2003; Krause et al., 2005). Thus, depending on the focus of a performance criteria, different parts of the hydrograph are emphasised or neglected (Pechlivanidis et al., 2014; Pfannerstill et al., 2014a; Haas et al., 2016).

In order to capture magnitude and dynamic in the modelled discharge time series, a combination of statistical performance metrics and signature measures in the model evaluation is recommended (van Werkhoven et al., 2008, 2009; Pechlivanidis et al., 2014; Pfannerstill et al., 2014a). An appropriate set of performance criteria should be selected so that all hydrological





conditions are represented by at least one performance criteria (Pfannerstill et al., 2014a). With increasing number of perfor-mance criteria more aspects of the hydrological behaviour are captured (Singh et al., 2011). Optimally, all performance criteria are complementary and related to different aspects of the hydrological system (Gupta et al., 1998; Clark et al., 2011; Pokhrel et al., 2012). The selection of an appropriate set of performance criteria is still challenging since the number and type of per-
formance criteria which are required to explain the hydrological behaviour in a study catchment is unclear and depends on the catchment characteristics and its underlying process complexity (Wagener and Montanari, 2011; Pokhrel et al., 2012).

For reliable identification of a model parameter, a performance criteria is appropriate which is related to the part of the hydrograph that is controlled by the selected model parameter. It is assumed that a performance criterion contributes to a better interpretation of the hydrological behaviour if it is related directly to the different components of the model structure
(Yilmaz et al., 2008; Gupta et al., 2009; Martinez and Gupta, 2010; Pechlivanidis et al., 2014; Sadegh and Vrugt, 2014). An implicit consideration of different hydrological components leads to a better understanding of the role of model parameters in controlling the hydrological behaviour in models (Guse et al., 2016a). It is thus intended to establish a strong relationship between a model parameter and a performance criterion which is appropriate for the associated process (Fenicia et al., 2007). Yilmaz et al. (2008) demonstrated that changes of model parameters in particular effect signature measures that are related to
the corresponding process such as that parameters controlling evapotranspiration are sensitive to low flows and water capacity parameters are more related to the bias between measured and modelled discharge time series.

Several studies showed that the relevance of model parameters changes when using different performance criteria (van Werkhoven et al., 2008; Abebe et al., 2010; Herman et al., 2013; Guse et al., 2014). Gupta et al. (2009) emphasised the need to investigate how changes in model parameter values influence the three components of the Kling-Gupta-Efficiency (KGE).
For a clear detection of model parameters, different performance criteria are required to determine whether parameters are only relevant specifically for a certain performance criteria. Moreover, it was shown that the relevance of model parameters is site-specific depending on the prevailing dominant processes (Gupta et al., 2014; Guse et al., 2016b). Thus, representative investigations of the relationship of model parameters and performance criteria requires analyses in contrasting catchments with differences in the dominant processes.

Performance measures were already classified in terms of catchment characteristics. Singh et al. (2014a) and Pechlivanidis and Arheimer (2015) classified performance criteria in different groups by using Classification and Regression Trees (CART) to identify the drivers of model performance. Both studies showed how different catchment characteristics and derived signatures result in a typical model performance. By doing so, the hydrological conditions for a specific model were detected which led to a good (poor) model performance. These studies presented an approach to explain the model performance based on catchment
characteristics.

This idea can transfered to the relationship between performance measures and model parameters. A better understanding of how model parameters and performance criteria are interrelated is a core idea of diagnostic model analysis (Gupta et al., 2008, 2009). For a precise parameter identification, the most relevant performance criteria for each model parameter needs to be derived. Thus, to our knowledge an explanation of how performance criteria are controlled by different model parameters
using regression trees and how this relationship changes for different types of performance criteria is still missing.





This study aims to identify the connective strength between model parameters and performance criteria. The connective strength assesses how strongly model parameters and performance criteria are interrelated using regression trees (RT). At first, it is investigated how the most influencing parameters vary for the selected performance criteria to analyse how strongly a set of model parameters affects different performance criteria. Second, looking from the side of the model parameters, it is analysed

which performance criteria are impacted by changes in a certain model parameter to detect which performance criteria are able to represent changes in a certain parameter. A high connective strength is given in the case that first a performance measure is controlled by one model parameter and second this model parameter influences the same performance measure to a relevant extent. This means that the model parameter has no relevant impact on other performance measures. This approach intends to contribute to an identification of an appropriate set of performance criteria as initial step for a precise identification of model

parameters.

## 2  Methods and Materials

### 2.1  Study catchments

In contrasting catchments, different hydrological processes are of major relevance (Atkinson et al., 2002; Merz and Blöschl, 2004; Jothityangkoon and Sivapalan, 2009; Guse et al., 2016b). Thus, also the ability of a certain performance criteria in

identifying a certain model parameter varies. With increasing relevance of a process, an accurate reproduction becomes more important. Thus, to check the applicability of the proposed approach, two catchments with different catchment characteristics are selected and measured daily discharge time series from their catchment outlets are used for this study.

### 2.1.1  Treene

The Treene catchment (up to the hydrological station Treia, 481 km$^2$) is as a typical lowland catchment as indicated by the

strong groundwater influence even under high flow conditions (Guse et al., 2014; Pfannerstill et al., 2014b; Guse et al., 2016b). Moreover, tile flow is a relevant process evoked by high drainage activities (Kiesel et al., 2010). Other fast runoff components are of minor relevance as it is expected from the low topographic gradient in the catchment (elevation only up to 80 meters). The Treene catchment is dominant by agricultural areas whilst only a minor part is covered by forests and urban areas (Guse et al., 2015).

### 2.1.2  Upper Saale

The Upper Saale catchment (hydrological station Blankenstein, about 1000 km$^2$) is located in the mid-range mountains of Germany. This catchment is characterised by a higher diversity in dominant processes compared to the Treene catchment with temporally changes in the relevance of snowmelt, surface runoff as well as groundwater flow as highlighted by Guse et al. (2016b). The landscape is covered mostly by forests (upper parts) and agriculture (lower parts). Compared to the Treene





catchment, altitude is higher (between 415 and 856 metres) and slopes are steeper. Thus fast runoff components are of higher relevance.

## 2.2 Soil and Water Assessment Tool (SWAT)

The conceptual and process-based eco-hydrological model SWAT (Soil and Water Assessment Tool, Arnold et al. (1998)) is

used in this study. The SWAT model is spatially discretised into subbasins which are sub-divided into hydrological response units (HRUs) based on unique information in landuse, soil and slope. The HRU is the central calculation unit for which the water balance is resolved. The change in soil water storage is calculated for each day as evoked by inputs (e.g. precipitation) and outputs (e.g. evapotranspiration, runoff components).

In this study, the SWAT3S-version (Pfannerstill et al., 2014a) as a further development of the SWAT model was used. In

SWAT3S, the groundwater modelling has been improved by subdividing the active aquifer contributing to the river discharge into a fast and a slow responding one. The SWAT model set-up for both catchments was realised as described in Guse et al. (2016b). For a detailed description of the model set-up, we refer to this study.

Twelve SWAT model parameters from different hydrological components are selected (Tab. 1) to analyse the relationship of performance criteria to model parameters which are controlling different parts of the hydrograph. The final selection is based

on studies with successfull applications of the SWAT model within the studied catchments (Guse et al., 2016b; Pfannerstill et al., 2015).

[Table 1 about here.]

Model simulations were carried out based on 2000 different parameter sets that were generated with the Latin Hypercube sampling approach as it is implemented in the r-package FME (Soetaert and Petzoldt, 2010). In the Latin Hypercube sampling,

all model parameters were changed simultaneously within the whole parameter space. For a more detailed description, please see Pfannerstill et al. (2014b).

## 2.3 Performance criteria

Ten performance criteria including five performance metrics and five signature measures were selected to capture different aspects of hydrological behaviour in models and as recommended in recent diagnostic model studies (Kling et al., 2012;

Pechlivanidis et al., 2014; Pfannerstill et al., 2014b; Haas et al., 2015)

The Nash-Sutcliffe Efficiency Criteria (NSE) (Eq. 1) is one of the most often used performance criteria in hydrology (Nash and Sutcliffe, 1970). The NSE focuses on the variability in the measured discharge time series. It is known to give higher weights to high flows than to low flows (Schaefli and Gupta, 2007; Gupta et al., 2009; Pfannerstill et al., 2014b).

$$NSE = 1 - \frac{\sum\limits_{i=1}^{N} (Q_o - Q_s)^2}{\sum\limits_{i=1}^{N} (Q_o - \bar{Q}_o)^2} \tag{1}$$





$Q_o$ is the measured discharge

$Q_s$ is the modelled discharge

$\bar{Q_o}$ is the mean of the measured discharge

The Kling-Gupta Efficiency criteria (KGE) (Gupta et al., 2009; Kling et al., 2012) is based on a decomposition of the NSE into its three components (Eq. 2), which can be separately considered for each model run. Model errors can be directly related to variability (KGE_alpha), bias (KGE_beta) and correlation (KGE_r) between measured and modelled discharge time series.

KGE_alpha is the variability ratio between the standard deviation of measured and modelled discharge values. A KGE_alpha larger than one shows that the variability in the modelled discharge time series is higher than in the measured discharge time series, while a lower KGE_alpha than one represents the opposite case. KGE_beta is the bias ratio between the average values for measured and modelled discharge. A KGE_beta larger than one represents an overestimation of discharge, i.e. a positive bias, while lower values than one illustrate an underestimation. KGE_beta and KGE_alpha represent the reproduction of the

first and second moments, respectively, as emphasised by Kling et al. (2012). KGE_r represents the correlation coefficient according to Pearson. Using KGE_r, the agreement in temporal dynamics between measured and modelled discharge time series is analysed. To calculate the KGE, the Euclidean distance to the ideal point in the three-dimensional criteria space which is created by its three components is calculated (Gupta et al., 2009). All three KGE components as well as the KGE have an ideal value of one.

$$KGE = 1 - \sqrt{(KGE\_alpha - 1)^2 + (KGE\_beta - 1)^2 + (KGE\_r - 1)^2} \qquad (2)$$

$KGE\_alpha \quad = \sigma_s/\sigma_o$

$KGE\_beta \quad = \mu_s/\mu_o$

$KGE\_r \quad$ = correlation coefficient

In addition to these five performance metrics, five signature measures are selected based on the FDC. The FDC only considers the discharge magnitude without considering the temporal occurrence of discharge values (Vogel and Fennessey, 1996; Yilmaz et al., 2008; Westerberg et al., 2011). To evaluate the model performance, the FDC is subdivided into five FDC segments

(very high (0-5% days of exceedance), high (5-20%), medium (20-70%), low (70-95%), very low (95-100%)) as proposed by Pfannerstill et al. (2014b). The FDC signatures consider that different discharge magnitudes are controlled by different processes. Whilst the high flow segment is mainly impacted by precipitation and fast runoff components, low flows are controlled by evapotranspiration and deep groundwater storages (Yilmaz et al., 2008; Cheng et al., 2012; Pokhrel et al., 2012; Yaeger et al., 2012; Guse et al., 2016b)

Each FDC segment was separately evaluated. Therefore, the ratio of the root mean square error to the standard deviation (RSR) was calculated for each FDC segment (Eq. 3) (Moriasi et al., 2007). The standardisation allows a fair comparison between the different segments (Haas et al., 2016). The optimal value for the RSR is zero. Using these five signature measures,




it can be derived how model parameters are related to the different discharge magnitudes (Pfannerstill et al., 2014b; Guse et al., 2016b).

$$RSR = \frac{\sqrt{\frac{1}{N} \sum\limits_{i=1}^{N} (Q_o - Q_s)^2}}{\sqrt{\frac{1}{N} \sum\limits_{i=1}^{N} (Q_o - \bar{Q}_o)^2}} \tag{3}$$

These ten different performance criteria were calculated for all 2000 simulation runs. Both, parameter sets and calculated performance criteria from these simulations were then used for the following analyses.

To analyse the relationship among the different performance criteria the correlation coefficients between all pairwise combinations were calculated. This correlation analysis enables the detection of (dis)similarities between performance criteria. Similarities in the performance criteria as indicated by a linear relationship in the dotty plots would show that these performance criteria capture a similar type of model error for this catchment. The intention was to detect whether each performance criterion provides additional information of the model error and whether the relationships were similar in both catchments.

### 2.4 Regression Trees

Regression Trees (RT) are a method to order the relationship between several explaining variables and a single target variable (Breiman et al., 1984). In a sequence of regressions, subsequently that variable is determined from a set of variables that has the highest predictive value for the target variable being analysed. An example of a regression tree will be shown in the results.

A regression tree is a binary algorithm using on a logical expression. In each step the data set is subdivided into two subsets (Singh et al., 2014b, a). The sequence of decisions is visualised in a tree diagram and allows to visually explore the importance of the different variables as predictor variables for the target variable. The (sub)set of model simulations is subdivided in each node of the tree into two groups defined by a threshold value for one of the explaining variables. All simulations with a value in the explaining variable above the threshold belong to the one group, and those with a value below the threshold to the other group. For each node, this approach is repeated until no further subdivision of a variable at a certain node explains the target variable. Either a different or an already chosen explaining variable is selected in the next branch of the tree. A regression tree consists of multiple branches depending on the complexity of the relationship between explaining and target variables. The earlier a variable is used in the construction of a RT, the higher is its importance. The variable used in the first split has thus the maximum importance. The gain in information is maximised by defining clearly separated subgroups of the whole simulation set (Singh et al., 2014b).

For our analyses, we used the R package rpart (Therneau and Atkinson, 2010). Using the RT algorithm, the contribution of each explaining variable on changes in the target variable is calculated. The percentage contribution of each explaining variable shows its importance for the target variable (Singh et al., 2014b).





### 2.4.1 Regression trees using performance criteria as target variable (RTperf)

Regression trees are applied in this study in two approaches using the 2000 model simulations with the pre-selected model parameters and the calculated performance criterion. In a first application the ten performance criteria are used consecutively as target variable to construct regression trees for each performance criteria (named RTperf). As explaining variables, the

model parameters are used to detect which model parameters lead to changes in a certain performance criteria. The relevance of each model parameter is derived from regression trees by calculating the percentage contribution of each model parameter in explaining the variability in a certain performance criterion. In this way, the most relevant model parameters are identified for each performance criterion.

### 2.4.2 Regression trees using model parameters as target variable (RTpar)

It is not only interesting to detect the most relevant parameters for a certain performance criterion. It is also important to know which performance criterion is most strongly impacted by changes in the values of a certain parameter. The latter point could not be derived from RTperf.

Thus, the next step was initialised vice-versa to RTperf from the side of model parameters to analyse how changes in model parameter influence the performance criteria. To achieve this, explaining and target variables in RT are permuted. Each model

parameter is used as target variable in RT and all performance criteria as explaining variables (named RTpar). In this way, it is detected which performance criterion is most strongly impacted by changes in the values of a certain model parameter. Thus in RTpar, the other model parameters are not directly used. Similarly to RTperf, the percentage contribution of each performance criterion is calculated to explain the impact of change in the values of a certain model parameter.

### 2.4.3 Connective strength by comparing both regression tree approaches

Subsequently, the percentage contributions as derived from both RT approaches are compared to analyse the connective strength between model parameters and performance criteria. We can differentiate into four cases of connective strength for each pair of model parameter and performance criteria (Fig. 1).

1. High percentage contributions in both RTs (RTperf, RTpar):

   Similar results of high percentage contributions in both RTs indicate a high bijective relationship between model pa-

rameter and performance criterion. In this case, the model parameter is clearly identifiable by the selected performance criterion. This is the optimal case representing a high connective strength and occurs if a certain parameter influences one performance criterion to a large extent without influencing other performance criteria significantly.

2. High percentage contribution in RTperf, but low in RTpar:

   In this case, a certain model parameter controls the selected performance criterion. However, this model parameter also

influences other performance criteria. This case occurs if the corresponding process is very dominant and influences multiple performance criteria. Here, the connective strength cannot be fully understood when using the performance





criteria as the target variable. From the side of the model parameter, it has further investigated which performance criterion is most appropriate for parameter identification.

3. Low percentage contribution in RTperf, but high in RTpar:

In this case, the model parameter is not the major controlling parameter on the selected performance criterion as detected in RTperf. However, its impact on other performance criteria is even lower which results in a high value in RTpar. Thus, the selected performance criterion is appropriate to explain the impact of changes in this model parameter, but the performance criterion is even stronger impacted by other model parameters. This case occurs if the corresponding process is of a minor relevance in describing the hydrological system of the catchment. Thus, due to the low process relevance, the connective strength is also low and a precise parameter identification is not found.

4. Low percentage contributions in both RTs:

In this case, neither this model parameter impacts the performance criterion to a relevant extent, nor the performance criterion is impacted by changes in the parameter. Thus, the connective strength is low. This parameter is not identifiable due to the low relevance of the corresponding process and a low relationship to one of the performance criteria.

[Figure 1 about here.]

By applying this approach to two catchments with different characteristics, it is analysed how strongly a certain performance criterion is connected to a specific model parameter and how this connective strength depends on the relevance of the corresponding process.

## 3 Results

### 3.1 Correlation between performance criteria

Pairwise correlation analysis for the performance criteria is carried out separately for each catchment. In the Treene catchment (Fig. 2, upper panel), NSE, KGE and the RSR of very high and high segments of the FDC are strongly correlated. Moreover, the RSR of low and very low flows are highly correlated. The KGE is mainly controlled by its variability component (KGE_alpha). A good performance of KGE_alpha (optimum=1) also results in a high performance in KGE. The KGE_beta (bias component) is correlated with the mid segment of the FDC. Concerning the values of the performance criteria, KGE_alpha and KGE_beta are mostly higher than one, indicating an overestimation and higher variability in the modelled discharge than in the measured. A good performance in a certain segment of the FDC occurs in the case of a good performance in the adjacent segment(s). In the case of a good performance for low flows also very low flows perform well. Similarly, also a good performance for very high flows was detected in model runs with a good performance in high flows. However, the correlations between the RSR of (very) high and (very) low flows are lower which indicates that there are less model runs with a good performance in both high and low flows.

[Figure 2 about here.]





In the Saale catchment (Fig. 2, lower panel), the correlations are overall lower. The strongest correlation is observed between NSE and the KGE_r. The KGE is correlated to KGE_alpha and KGE_r. Thus, both variability and correlation in the modelled discharge time series are relevant for a good performance of the KGE. The KGE_beta in contrast, which is balanced between over- and underestimation, is of lower relevance. The correlation among the signatures of the FDC segments is lower compared to the Treene catchment, even between adjacent segments. Here, a good performance of low flows does not result in a good performance of very low flows. A worse performance of KGE-alpha (higher or lower than one) leads to a decrease in the KGE since it increases the Euclidean distance of the three KGE components. However, as shown in Fig. 2, a different result was obtained between KGE_alpha and NSE. A lower variability in the modelled than in the measured discharge time series (KGE_alpha<1), results in an improvement of the NSE. In contrast, a KGE_alpha larger than one indicates that the variability is higher in the modelled discharge time series which leads to a reduction of NSE. This corresponds with the calculation of NSE which strongly emphasises the variability in the measured time series.

### 3.2 Impact of model parameters on performance criteria (RTperf)

The connective strength between model parameters and performance criteria was investigated using regression trees (RT). At first, RTs were constructed using the ten performance criteria (RTperf) as target variables. The aim of this step was to detect which model parameter most strongly affects a certain performance criterion.

Fig. 3 (above) shows the regression tree for the KGE for the Treene catchment exemplarily for RTperf. Looking from top to down, the most influencing model parameters for the KGE are provided. The first branch is defined by the groundwater retention time of the first aquifer (GW_DELAYfsh) and the second one on the one right side again by GW_DELAYfsh and on the left side by the aquifer partitioning coefficient (RCHRGssh). In total, only groundwater parameters affect the KGE to a relevant extent. When going along the branch at the right side, parameter settings of the controlling model parameter at these nodes are identified which lead to the best KGE on average (0.83).

[Figure 3 about here.]

To assess the connective strength between model parameters and performance criteria, the percentage contribution of model parameters as explaining variables for each performance criterion is shown for both catchments (Fig. 4). The parameter contribution in the Treene catchment to explain the variability in the performance criteria can be classified into three groups (Fig. 4, above). At first, six performance criteria are mainly influenced by GW_DELAYfsh and to a lower extent by RCHRGssh which shows the strong dominance of groundwater processes (see also Guse et al. (2014)). However, since multiple performance criteria are influenced by GW_DELAYfsh and RCHRGssh, the most appropriate performance criterion to identify the impact of these parameter is not detectable. The second group consists of KGE_beta and the RSR of the mid flow FDC segment. Both are controlled strongly by soil evaporation (ESCO) and available soil water capacity (SOL_AWC). Thirdly, low and very low flows are controlled in addition to GW_DELAYfsh by the baseflow recession coefficient of the second aquifer (ALPHA_BFssh). Seven of the twelve model parameters namely fast runoff (CN2, SURLAG, GDRAIN, LATTIME),



soil (SOL_K) and snow parameters (SFTMP, SMTMP) have only a minor impact on all performance criteria and cannot be identified by the selected performance criterion.

[Figure 4 about here.]

Fig. 4 (below) shows that the relationship between model parameters and performance criteria is more complex in the Saale catchment compared to the Treene. A clear classification into groups of performance criteria which are controlled by certain model parameters is more difficult. Four performance criteria (KGE_alpha, KGE_r, NSE, very high flow segment of the FDC) are controlled by the lateral flow lag time (LATTIME). But these performance criteria are also influenced by groundwater parameters (GW_DELAYfsh, RCHRGssh). Furthermore, the RSR for high flows is not controlled by LATTIME but by these two groundwater parameters and the hydraulic conductivity in the soil (SOL_K). The KGE is controlled by a parameter (GW_DELAYfsh) which has not the largest percentage contribution for one of its three components. The water balance (KGE_beta) is controlled by ESCO and SOL_AWC, while mid flows are mainly influenced by SOL_AWC. Low flows are controlled by GW_DELAYfsh and very low flows by ALPHA_BFssh. Snow and fast runoff parameters except of LATTIME do not influence one of the performance criteria to a high extent. Thus, also in the Saale catchment, parameters exist without significant impact, while LATTIME controls multiple performance measures.

By detecting the controlling model parameters for each performance criterion, in both catchments no appropriate performance criteria are found for several model parameters (e.g. CN2) which shows the low connective strength between these model parameters and the performance criteria. Due to that, the identification of parameter values would be difficult. It is of importance to detect whether the low relevance of a model parameter is related to a minor relevance of the corresponding process or whether the selected performance criterion are inappropriate to identify this model parameter. Moreover, some model parameters have the highest influence on multiple performance criteria (e.g. GW_DELAYfsh and RCHRGssh in the Treene, LATTIME in the Saale), which leads to unclear results in the connective strength between model parameters and performance criteria. This suggests that these parameters extremely govern the overall hydrological system in the model.

### 3.3 Impact of changes in model parameters on performance criteria (RTpar)

In the second RT step, the relationship between model parameters and performance criteria is analysed using the 12 model parameters consecutively as target variables. It is investigated which performance criteria are impacted by changes in the model parameters (RTpar, Fig. 5).

Fig. 3 (below) shows the regression trees examplarily for the model parameter GW_DELAYfsh for the RTpar approach in the Treene catchment. Here, KGE_alpha separates the data set at the first node and occurs once at the two following branches. Moreover contrasting performance criteria are included (KGE_r, RSR for very high flows and very low flows).

In the Treene catchment (Fig. 5, above), the curve number (CN2) is most significantly related to the RSR for very high flows and furthermore to NSE, KGE and the RSR for high flows. This shows that CN2 and thus surface runoff controls the highest flow conditions. Snow parameters (SFTMP, SMTMP), the timing parameters for surface runoff (SURLAG) and tile flow (GDRAIN) as well as soil hydraulic conductivity (SOL_K) have the highest influence on KGE_r (correlation). Thus, variations





in these parameters lead to changes in the correlation between measured and modelled discharge time series. Concerning the soil model component, SOL_AWC and ESCO are strongly related to water balance (KGE_beta) and to a lower extent to RSR for mid flows. ALPHA_BFssh is related to the RSR of low and very low flows as well as to the NSE. For the two groundwater parameters (GW_DELAYfsh, RCHRGssh), four performance criteria (NSE, KGE, KGE_alpha, RSR for high flow) have a

similar percentage contribution. This point shows that both groundwater parameters control different aspects of hydrological model behaviour without having a clear relationship to a certain part of the hydrograph.

[Figure 5 about here.]

In the Saale catchment (Fig. 5, below), snow parameters (SFTMP, SMTMP) as well as LATTIME affect both KGE_r and NSE. Changes in the curve number (CN2) mainly influence KGE_alpha and the RSR for very high flows. Thus, the variability

between measured and modelled discharge time series and in particular high flows are influenced by CN2. All three soil parameters (SOL_AWC, SOL_K, ESCO) influence the water balance (KGE_beta) and the mid flow segment of the FDC. However, evaporation (ESCO) is more related to KGE_beta while SOL_AWC has the largest impact on mid flows. In the case of GW_DELAYfsh several performance criteria are affected to a similar extent, but none of them has a high percentage contribution. While RCHRGssh affects KGE and high flows, ALPHA_BFssh strongly controls very low flows.

### 3.4 Comparing RTperf and RTpar

Subsequently, both RT approaches are compared by relating the percentage contribution from RTperf to RTpar and analysing these patterns for each performance criterion for both catchments (Fig. 6).

[Figure 6 about here.]

For mid and low flow conditions, both RTperf and RTpar provide a strong connective strength with high percentage contri-

bution in RTperf and RTpar for the same pair of model parameter and performance criteria in both catchments. The strong relationship of evaporation (ESCO) and available soil water capacity (SOL_AWC) to the RSR of the mid flows and the KGE_beta is derived in both RT approaches (Fig. 6). The water balance (KGE_beta) is hereby more controlled by ESCO, whilst SOL_AWC is the dominant parameter for mid flows especially in the Saale catchment.

Similarly, the connection between the RSR for the very low segment of the FDC and the baseflow recession coefficient

(ALPHA_BFssh) is strong in particular in the Saale catchment. In both catchments, also the retention time of recharge into the groundwater (GW_DELAYfsh) is of relevance.

The KGE is dominated by GW_DELAYfsh and the aquifer partitioning coefficient (RCHRGssh) in a similar way in both catchments despite of contrasting catchment characteristics. The KGE is most strongly impacted by GW_DELAYfsh (see RTperf). However, in RTpar, the KGE has a higher percentage contribution in explaining the changes in RCHRGssh than in

GW_DELAYfsh.

In contrast, the performance criteria related to high flows (NSE, very high segment) are controlled in the Treene catchment by groundwater (GW_DELAYfsh, RCHRGssh) and in the Saale catchment by lateral flow (LATTIME). This pattern shows that





the NSE focuses on the model error at high flows. A lower connective strength between model parameters and performance criteria was detected for high flow conditions. A bijective relationship between high flow related performance criteria and certain model parameters is more difficult to detect. The five performance criteria representing high flow conditions in the Treene catchment are related to the same two groundwater parameters (GWDELAYfsh and RCHRGssh). However, whilst

GWDELAYfsh and RCHRGssh are the most dominant model parameters in RTperf, the percentage contribution in RTpar is lower. These two parameters dominate five performance criteria, but it remains unclear which is the best performance criterion in terms of parameter identification. Thus, while model errors in mid and low flows are identified by the same performance criterion (Case 1, see Chapter 3.3.3), it is more complex to find appropriate performance criteria for errors in high flows. Here, a more complex hydrological behaviour is in particular detected in the Saale catchment as indicated by different controlling

parameters on the performance criterion (Case 2 and 3 in Chapter 3.3.3). Moreover, the most dominant parameters in both catchments (GW_DELAYfsh in the Treene, LATTIME in the Saale) have a high percentage contribution in particular in RTperf both for high and low flows.

    A very specific pattern is detected for KGE_r in the Treene catchment. In RTpar, a high percentage contribution of KGE_r for model parameters of lower relevance is detected. The high values for RTpar and the low values for RTperf in Fig. 6 shows

that KGE_r is the most appropriate performance criterion to assess changes in these model parameters. However, due to the low relevance of snow or surface runoff, the KGE_r is controlled by groundwater parameters. Here, we see a large difference in the interrelationship between model parameters and performance criteria by comparing RTperf and RTpar.

## 4   Discussion

In this study, the connective strength between model parameters and performance criteria was investigated using two ap-

proaches of regression trees.

    The RT approach using performance criteria as target variables (RTperf) (see Fig. 4) shows that not all model parameters influence one of the selected performance criteria. However, using model parameters as target variables in RTpar it was detected which performance criteria are impacted by changes in the value of a model parameter. In this way, it can be derived whether the parameters and their associated processes are of low relevance or whether an appropriate performance criterion for a model

parameter is missing. The RTpar approach shows for the majority of the model parameters that the changes in their values are detectable at least by one of the selected performance criteria. Thus, the impact of parameters from processes of minor relevance on performance criteria can be derived in RTpar. This demonstrates that the impact of performance criteria depends on the relevance of the corresponding process (Sadegh and Vrugt, 2014). In this way, it can be derived whether the parameters and their associated processes are of low relevance or whether an appropriate performance criterion for a model parameter is

missing. In the case that the relevance of the associated process is very low, parameters from other more dominant processes control the performance criteria. This is for example shown for the curve number (CN2). Its impact on the performance criteria in the Treene catchment is low due to the higher contribution of groundwater flow compared to surface runoff.





Comparing the results of the two RT approaches, a higher similarity between both catchments is detected in RTpar (Fig. 5). Differences between RTperf and RTpar are also obtained for parameters related to the most dominant process(es). The groundwater parameters (mainly GW_DELAYfsh) control most of the performance criteria for the Treene catchment (Fig. 4). A similar result is obtained for the Saale catchment with a dominance of the lateral flow lag time (LATTIME).

The differences in the relevance of model parameters on the ten performance criteria emphasise the benefit of the use of this set of performance criteria. The separate consideration of KGE components demonstrates that different parameters are related to these three performance metrics. While the relevant parameters on KGE and KGE_alpha are similar in the Treene catchment, the most relevant parameter on the KGE in the Saale catchment (GW_DELAY) is not the relevant one for the three

KGE components. Since each KGE component can be clearly related to a specific part of the hydrological behaviour (Kling et al., 2012), the regression tree shows whether a model parameter is more relevant in representing the variability, correlation or bias in the modelled discharge time series. By comparing the KGE and its components, the most important aspect of the hydrologic behaviour for a good model performance becomes apparent such as the variability as detected by KGE_alpha in the Treene catchment.

The differences in pairwise correlations of the performance criteria between both catchments also results in differences in the controlling model parameters on a certain performance criterion. Similar results in KGE and NSE are calculated in the case of high correlation between KGE and KGE_alpha and thus the most relevant model parameters on these two performance criteria are similar. The opposite result is obtained in the Saale catchment. Due to the low values of the KGE_alpha, different parameters control KGE and NSE.

Concerning the signature measures, this study shows that different parameters are related to the FDC segments. This is in line with studies stating that each FDC segment can be related to certain catchment processes (Yilmaz et al., 2008; Yaeger et al., 2012). The strong connective strength of model parameters regulating water balance to mid flows as well as of parameters from slow reacting aquifer storages to very low flows is derived in this study. However, a typical sequence of a high connective strength of high flows to surface runoff parameters is not identified. Moreover, we can see that the dominant process, i.e.

groundwater flow in the Treene and lateral flow in the Saale catchment, influences both high and low flows. This leads to a trade-off in parameter identification since the same model parameters control high and low flows.

Our RT analysis in the two catchments shows that the most appropriate performance criterion varies depending on the different model parameters. The connective strength between pairs of model parameter and performance criteria varies as it

was detected in linking RTperf and RTpar. Pairs with a high connective strength were detected and grouped. This results in a minimum number of required performance criteria of at least three performance criteria related to high, mid and low flow conditions since the most relevant parameters between these three types of performance criteria vary. This is in line with other studies on performance criteria stating that 3-4 performance criteria capturing different parts of the hydrological system are a minimum number (Madsen, 2000; Boyle et al., 2001; van Werkhoven et al., 2008, 2009).



The need for an individual performance criterion to assess mid flows was mentioned as a good representative for the water balance (van Werkhoven et al., 2008; Wagener et al., 2009; Herman et al., 2013). The RT analysis shows that the controlling parameters for KGE_beta and the RSR for mid flows are different compared to the other performance criteria and these dominant model parameters are from the soil components and related to the water balance. In these cases, the connective strength is very high.

Also for an assessment of low flow conditions an individual performance criterion is needed which was in this case the RSR for low and very low flows. A high connective strength between model parameter and performance criterion was detected for very low flows. Moreover, the requirement for a segmentation of the FDC into very low and low flows as introduced by Pfannerstill et al. (2014a) is emphasised by identifying different relevant model parameters. The high correlation between the RSR for very low and low flows in the Treene catchment also results in similar dominant parameters, while different parameters control these signature measures in the Saale catchment (see Fig. 4). Thus, similar influencing parameters in RTperf for different performance criteria are detected if both are highly correlated.

On the other side, high flows are more complex and driven by interacting and overlaying processes from different hydrological components. Here, the most influential parameters and the most appropriate performance criterion vary depending on the type of errors which are dominant in the modeling process. The complexity in the representation of high flows depends on the involved processes. In the groundwater dominated Treene catchment, the RT analysis for the five performance criteria related to high flows provides very similar results. The RT analysis using the model parameters as explaining variables (RTpar) however highlights the differences in the relationship of model parameters and these five performance criteria. In the Saale catchment, the relevant model parameters largely vary between all performance criteria related to high flows. Here, all selected performance criteria capture different types of errors in the modelled discharge time series. The analysis of deviations between measured and modelled discharge in this catchment is more complex so that more than one performance criterion for high flows is required. With higher heterogeneity in dominant processes and strong interaction of different processes in controlling the hydrological behaviour, a more distinct selection of a larger set of performance measures is required.

Thus, it can be recommended to include several performance criteria to capture all types of potential errors both in dynamic and magnitude of the modelled discharge. In addition, it is relevant to consider which model parameters dominate the performance criteria also by comparing the dominant parameters between different performance criteria. This can help to understand why a certain model error might occur and to which processes this model error is related.

As demonstrated in this study, the results vary between different catchments. Further studies in other catchments might provide a more precise understanding of the connective strength between model parameters and performance measures. Due to the general methodological approach, an applicability with other models is expected.

## 5 Conclusion

For achieving a precise parameter identification, the connective strength between model parameters and different performance criteria is analysed. For this, two regression tree (RT) approaches are applied using consecutively the performance criteria and





the model parameters as target variables. This method derives which model parameters affect a performance criterion (RTperf) and which performance criteria are impacted by changes in the model parameters (RTpar).

The main outcomes are:

1. The pairwise correlation between performance criteria varies between the two catchments depending on the model error. Thus, different performance criteria are required to disentangle the impact of different hydrological behaviour on the modelled discharge. The number of required performance criteria is higher for catchments with a higher process complexity.

2. In RTperf, it becomes apparent how largely the relevance of model parameters varies between different performance
criteria. Our study emphasises the relevance of a separate consideration of the KGE components and of a signature-based analysis of different FDC segments for a precise parameter identification. Differences in the dominant parameters are detected between performance criteria related to high, mid or low flow conditions, respectively.

3. RTpar using the model parameters as target variables shows which performance criterion is appropriate to identify a model parameter. Similar results in RTperf and RTpar demonstrate the high capability of a performance criterion to
consider the impact of a model parameter accurately. Contrasting results are in particular derived for model parameters which are related to processes of minor relevance. A bijective connective strength between model parameters and performance criteria is detected for low and mid flows, whilst the modelling of high flows is more complex both in terms of relevant model parameters and appropriate performance criteria.

Overall, this study shows that multiple performance criteria are required for an accurate parameter identification for re-
liable hydrological modelling. The connective strength between model parameters and performance criteria varies between catchments depending on the hydrological complexity of the catchments with respect to the processes and their relevance in controlling the hydrological behaviour in models. Using our approach, it can be derived how precisely model parameters can be identified by a set of performance criteria.

*Acknowledgements.* We thank the DFG for financial support of the first author (project GU 1466/1-1 Hydrological consistency in modelling).
Furthermore we thank the CAWa (Central Asian Water) project (www.cawa-project.net, Contract No. AA7090002), by the German Federal Foreign Office as part of the German Water Initiative for Central Asia ("Berlin Process") for the funding of the third author (AG). The fourth author (JK) was funded through the "GLANCE" project (Global change effects in river ecosystems; 01LN1320A) supported by the German Federal Ministry of Education and Research (BMBF). The fifth author (CL) received funding by the Leibniz Association (SAW-2012-IGB-4167) within the international Leibniz graduate school: Aquatic boundaries and linkages in a changing environment (Aqualink)
(http://www.igb-berlin.de/aqualink.html).
We thank Schleswig-Holstein Agency for Coastal Defence, National Park and Marine Conservation of Schleswig-Holstein (LKN-SH) and the State Institute for Environment and Geology of Thuringia (TLUG) for the discharge data. We thank Martin Volk and Michael Strauch



(UFZ) for contributing to the SWAT modelling in the Saale catchment. Furthermore, we thank the community of the open source software R, which we used for this study.



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





## List of Figures




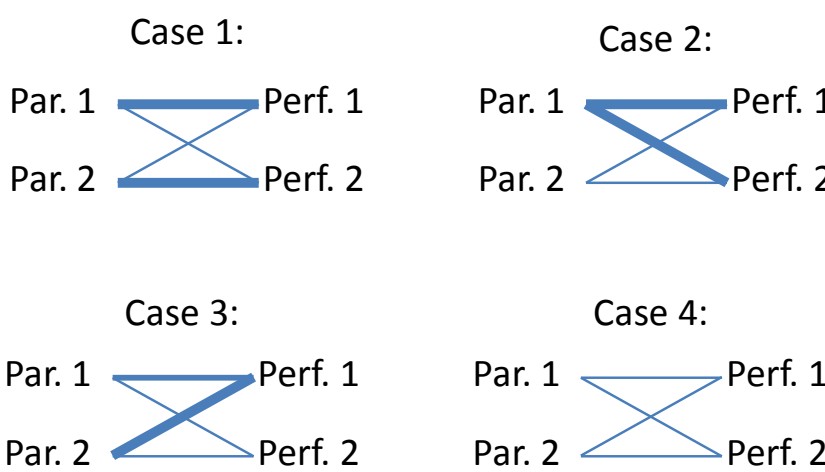

**Figure 1.** Flowchart of four cases of connective strength between model parameters and performance measures. A wider blue line shows a higher impact of the model parameter on the performance criteria





**Figure 2.** Scatterplot matrix of the performance criteria of the Treene (in black, upper panel) and Saale (in grey, lower panel) catchment showing pairwise performance criteria plots for the 2000 model simulations. The scales on the sides show the values of the respective performance criterion.







**Figure 3.** Example of a regression tree (RT) using (above) KGE as target variable and model parameters as explaining variables and (below) the model parameter GW_DELAYfsh as target variable and performance measures as explaining variables for the Treene catchment.





**Figure 4.** Regression trees (RT) using performance criteria as target variables. The percentage contribution of the model parameters in explaining the performance criteria is shown for the Treene (above) and Saale (below) catchment. In every row the percentage contributions sum up to 100%.







**Figure 5.** Regression trees (RT) using model parameters as target variables (RTpar) for the Treene (above) and Saale (below) catchment and performance criteria as explaining variables. All values of a parameter are in white in the case that the resulting variation among the performance criteria for this parameter was too low to construct a regression tree. In every row the percentage contributions sum up to 100%.





**Figure 6.** Connective strength between performance criteria and model parameters. The percent contribution of the pairs of model parameter and performance criterion are shown as derived from RTperf (x-axis) and RTpar (y-axis). A high value along the x-axis shows a high contribution of a model parameter in explaining the variability in the performance criterion as detected by RTperf. A high value along the y-axis (RTpar) shows that this performance criterion among all performance criteria is most strongly impacted by changes in the model parameter. A strong connective strength is detected if both values are high. The pairs with at least one high percent contribution are labeled. The results from the Treene catchment are shown as black circles and from the Saale catchment in grey squares. Please note that percentage contributions on the x-axis sum up to 100%, while this is not the case for the y-axis.





**List of Tables**





**Table 1.** List of SWAT models parameters. Lower and upper ranges are given as absolute range (r), additive (a) or multiplicative (m) value. Further information can be found in the theoretical documentation of the SWAT model (Neitsch et al., 2011).

| Parameter name | Abbreviation | Process | Range type | Lower range | Upper range |
|---|---|---|---|---|---|
| Snow fall temperature | SFTMP | Snow | r | -2.5 | 2.5 |
| Snow melt temperature | SMTMP | Snow | r | -2.5 | 2.5 |
| Curve number | CN2 | Surface runoff | a | -10 | 10 |
| Surface runoff lag time | SURLAG | Surface runoff | r | 0.8 | 4 |
| Lateral flow lag time | LATTIME | Lateral flow | r | 0.2 | 8 |
| Tile flow lag time | GDRAIN | Tile flow | m | 0.5 | 1.5 |
| Available water capacity of a soil layer | SOL_AWC | Soil water | a | -0.02 | 0.1 |
| Saturated hydraulic conductivity of a soil layer | SOL_K | Soil water | m | 0.5 | 3 |
| Soil evaporation compensation factor | ESCO | Evapotranspiration | r | 0.2 | 1 |
| Groundwater delay time (fast aquifer) | GW_DELAYfsh | Groundwater | r | 1 | 50 |
| Aquifer fraction coefficient (slow aquifer) | RCHRGssh | Groundwater | r | 0.2 | 0.8 |
| Baseflow alpha factor (slow aquifer) | ALPHA_BFssh | Groundwater | r | 0.001 | 0.2 |