# Peer review of "Identifying the connective strength between model parameters and performance criteria"

_Hydrology and Earth System Sciences, 2017_

## Referee Comment (RC1) · Anonymous Referee #1 · 12 Mar 2017

**Identifying the connective strength between model parameters and performance criteria By Guse et al.**

**General Comments**

This study presents an approach to quantify the strength of the bijective relationship between model parameters to performance measures. The proposed method explores the model parameter-performance space using regression trees with the goal to detect performance measures that can uniquely identify a parameter. The regression trees are first developed by casting model parameter as explanatory variables and performance measure as prediction variables, and then by exchanging the explanatory and prediction variables. These trees are developed for two catchments in Germany.

The main idea presented in the study is interesting and results contribute valuable insights towards model diagnostics. However, there are a few issues that should be addressed. First, the introduction requires revision so as to remove repetition of ideas (see for example, lines 11 and 32 on Page 2), and to provide more background. The need for multiple performance measures to identify unique aspects of the hydrograph is well motivated, but why this has remained a challenge is not discussed. Some well-known issues are parameter interaction, limited information content in hydrologic time series data that allows identification of only a handful of parameters, and uncertainties in input as well as streamflow data (Beven, 2011). Complicating this further is the time varying nature of parameter sensitivity (Herman et al. 2013).

Another important issue is the comparison of the proposed method to already existing sensitivity analysis methods, which also attempt to identify relationship between model performance and parameters. In my understanding, it is the partitioning of the model performance space by parameter values that is unique about regression trees, but the order of importance of parameters should ideally be the same as that derived using sensitivity analysis methods.

**Specific Comments**

1. Method description: Further information on implementation of regression trees is warranted. For example, the metric: 'percentage contribution of each explaining variable' is used throughout the manuscript without an explanation to how it is actually estimated by regression trees. It is expected that any method will have some error or uncertainty associated with its results, so what levels of 'percentage contribution' are significant? If any cross validation analysis during tree construction was used, it should be explained.

2. Methodological choices: It is mentioned in the manuscript that it is likely some performance measures are correlated (Lines 6-9, Page 7), these correlations are also presented in Figure 2. However, all performance measures are used for generating trees for RTpar. Could this potentially be the reason behind poor connective strength between performance measures and parameters for high flows? It should be discussed whether the regression tree algorithm can deal with correlated input. If not, correlated performance measures should ideally be reduced to an uncorrelated set. In fact, the

same holds true for the use of model parameters as independent variables in RTperf, the presence of parameter interaction will affect the results to some extent.

3. Background data: The time period of analysis, values of catchment average precipitation, temperature, etc. should be provided. An appendix with some details on the model structure and implementation of SWAT can be considered to make the study independent of prior applications of the model to these catchments. As the main focus is model diagnostics, it is essential that readers are aware of the model structure and the details of its implementation.

4. Issue of CN2 (Line 16, Page 11): It is surprising that no appropriate performance criteria is found to relate to CN2, which is generally a sensitive parameter in SWAT. One reason can be the low variation assigned to it (only within +/- 10 of base value, see Table 1). On the other hand, some other parameters are allowed to vary within much larger ranges (GW_DELAYfsh between 1-50, RCHRGssh between 0.2-0.8, etc.). It is later found that these parameter display high connectivity to performance measures. Please also mention the units of parameters in Table 1.

5. Threshold of performance: Figure 1 shows that negative NSE and KGE values were also allowed in the tree construction. The issue of using parameter sets related to highly degraded performance has been raised and addressed by earlier studies (Kelleher et al. 2013). Should a threshold of performance be fixed and only those parameter sets that perform above it considered for further analysis?

6. Convergence of results with number of LHS samples: 2000 parameter sets are used in the analysis but no discussion on the stability of results w.r.t number of LHS samples is provided. One way to test this is to look at the agreement between current results with those from a subset of 500, and 1000 sets. Typically, the number of sets after which little fluctuation in results is seen is used.

7. Equation 3, Page 7: Please elaborate how the RSR calculation is implemented. Say there are only 10 flow values for 0-5 percentile range for observed flow but 100 such values are present for simulated flow, how is RSR then calculated?

**Technical Corrections**
1. Line 1, Page 1: Consider replacing 'parameters are used to adapt the model to the conditions of the catchment' with 'parameters are used to represent the time-invarying characteristics of the catchments'.
2. Line 1, Page 2: Consider replacing 'In models' with 'In rainfall runoff models'.
3. Lines 7-9, Page 2: It is now generally accepted that parameters may or may not be identifiable (Beven, 2011).
4. Line 21, Page 4: Explain 'high drainage activities'.
5. Line 28, Page 4: Replace 'temporally' with 'temporal'.
6. Line 22, Page 11: Remove 'extremely'.

7. The text size in Figure 3 (lower panel) should be increased for visibility.

**References**

Beven, K.J., 2011. Rainfall-runoff modelling: the primer. John Wiley & Sons.

Herman, J.D., Reed, P.M. and Wagener, T., 2013. Time-varying sensitivity analysis clarifies the effects of watershed model formulation on model behavior. Water Resources Research, 49(3), pp.1400-1414.

Kelleher, C., Wagener, T., McGlynn, B., Ward, A.S., Gooseff, M.N. and Payn, R.A., 2013. Identifiability of transient storage model parameters along a mountain stream. Water Resources Research, 49(9), pp.5290-5306.

---

## Referee Comment (RC2) · Anonymous Referee #2 · 22 Mar 2017

In this manuscript, parameter uncertainty (incorporated with the SWAT model) was explored by using Latin Hypercube sampling. In general, the manuscript is overall well-written and I personally really like Figure 4 and 5 (the way of presentation). However, I cannot recommend for publication in HESS for mostly the reason of novelty and also the following concerns: 1.Parameter uncertainty along with complex watershed models (in this case, SWAT, or other cases such as HSPF, MIKE SHE, and others) has been extensively explored for decades. It does not mean there's no value (in terms of academic novelty) in investigating parameter uncertainty anymore, however, similar approaches (parameter uncertainty, sensitivity, model calibration for flow related variables) have been conducted previously. The proposed work may not meet the scientific standards of HESS. The value of this work may be enhanced by highlighting some local issues such as (i) what's the current concern(s) (Agricultural? Domestic? Industrial?

Environmental?) in the Treene and Upper Saale catchments; and (ii) what would be the benefit(s) to use the propose approach in the study area. 2.Details of both catchments were not provided. I would say most people know the location of Germany but maybe not the given two catchments. Parameter uncertainty and the associated comparisons may not be very much meaningful if the information of the targeted regions was not clear. 3.It seems that previous work from H. V. Gupta (famous scholar we know that), B. Guse, and M. Pfannerstill was cited a lot in the manuscript. However, as I mentioned previously, there are many other similar research available (at least in the past 10∼15 years) but not being discussed or compared. It also may be a considerable issue of the proposed work for the general evaluation against others was not provided.

---

## Author Comment (AC1) · 27 Mar 2017

Comment from reviewer#2: In this manuscript, parameter uncertainty (incorporated with the SWAT model) was explored by using Latin Hypercube sampling.

Reply from the authors: We have to mention that we do not agree with this summary of our manuscript. As clearly stated in the manuscript, our focus was to investigate the relationship between model parameters and performance measures. Thus, we show which model parameters impact which performance measure and which performance measures are influenced by the different model parameters. This approach is fundamentally different from a parameter uncertainty analysis. We do not tangle the parameter values in the manuscript and did not investigate the uncertainty of model parameters. We even do not mention the term "uncertainty" in the entire manuscript.

Based on the overall summary of the reviewer we suspect a severe misunderstanding of the topic of our study. We kindly ask the reviewer to clarify his statement considering our hopefully clarifying comments. In particular, we would appreciate if the reviewer could clarify why he came to the conclusion that this manuscript deals with a study of a parameter uncertainty analysis.

C: In general, the manuscript is overall well written and I personally really like Figure 4 and 5 (the way of presentation).

R: We thank the reviewer for this very positive comment.

C: However, I cannot recommend for publication in HESS for mostly the reason of novelty and also the following concerns:

R: To our knowledge, the idea of investigating the relationship between model parameters and performance measures from both sides is certainly new. This is also worked out in the introduction of the manuscript. Based on our literature review, we propose the concept of connective strength which was introduced in this manuscript and is to our knowledge new to the hydrological modelling community. Moreover, we think that it is still a challenge to understand which performance measures are really able to capture the variation in a certain model parameter and how this relationship varies for different catchments. If a set of performance measures is used in a hydrological study, it is still unclear which model parameters are adequately captured by at least one of the performance measures and which model parameters are not identifiable at all by the selected performance measures. Moreover, we show how many model parameters are impacted by a certain performance measure. In case of high bijective connective strength the model parameter can be clearly identified with the respective performance measure. If several performance measures influence the same model parameter(s) or one performance measure is impacted by a set of model parameter, the parameter identification is limited compared to the first case. For all these reasons we argue that our manuscript provides indeed new ideas, which might be beneficial for the hydrological community.

C: 1. Parameter uncertainty along with complex watershed models (in this case, SWAT, or other cases such as HSPF, MIKE SHE, and others) has been extensively explored for decades.

R: We agree that there are several studies on parameter uncertainty, but as mentioned above, this was not the topic of the current manuscript. According to our previous explanations, our intention is to improve the parameter identification of a hydrological model with a new methodical approach that fundamentally differs from uncertainty analysis.

C: It does not mean there's no value (in terms of academic novelty) in investigating parameter uncertainty anymore, however, similar approaches (parameter uncertainty, sensitivity, model calibration for flow related variables) have been conducted previously.

R: As we have argued above, the concept of connective strength and investigation of the relationship between model parameters and performance measures is new. We are not aware of any closely related paper. We would be thankful if the reviewer could provide examples, if available.

C: The proposed work may not meet the scientific standards of HESS.

R: We would be glad if the reviewer could give some more information why the scientific standard of HESS is not fulfilled. We think that this is something different from the argument that there are already similar studies (which is, in our opinion, not the case).

C: The value of this work may be enhanced by highlighting some local issues such as (i) what's the current concern(s) (Agricultural? Domestic? Industrial? Environmental?) in the Treene and Upper Saale catchments; and (ii) what would be the benefit(s) to use the propose approach in the study area.

R: We have selected both catchments since they are characterised by different landscapes (lowland vs upland). As suggested by the reviewer, we will provide additional

information on the catchments in the revised version of the manuscript. We think that these contrasting catchments are appropriate to show how the results change between the catchments. Our approach is in general applicable to all models and catchments, since the core idea is to analyse the relationship between model parameters and performance measures. This is of general relevance in all model applications.

C: 2. Details of both catchments were not provided. I would say most people know the location of Germany but maybe not the given two catchments.

R: We agree that we could give more information on the catchments including their location and we will do so in the revised version of the manuscript.

C: Parameter uncertainty and the associated comparisons may not be very much meaningful if the information of the targeted regions was not clear.

R: We hope that this point becomes clearer after improving the presentation of the catchments as outlined above.

C: 3. It seems that previous work from H. V. Gupta (famous scholar we know that), B. Guse, and M. Pfannerstill was cited a lot in the manuscript.

R: Since this manuscript is based on former studies of these three authors, we had to include some of their papers. However, we can try to reduce the number. In this context, we like to emphasise that we included more than one paper from other first-authors as well such as T. Wagener (4 times), K. Van Werkhoven and R. Singh, since our work is also based on their studies. We would like to emphasise that we did not have any joined publication with them. Thus, overall, we think that the reference list is appropriate and fairly balanced to meet the requirements of the scientific standard.

C: However, as I mentioned previously, there are many other similar research available (at least in the past 10_15 years) but not being discussed or compared.

R: As suggested by the reviewer, we can enhance the discussion by including more papers from past on the overall topic. However, again we would like to emphasize that

parameter uncertainty is not the topic of this manuscript so that the number of papers dealing with parameter identification is very limited and is already integrated.

C: It also may be a considerable issue of the proposed work for the general evaluation against others was not provided.

R: We do not understand which kind of comparison is expected here. We kindly ask the reviewer to give some more detailed explanation so that we can consider a possible comparison in a revised version of the manuscript.

---

## Editor Comment (EC1) · D. Solomatine (Editor) · 28 Mar 2017

The authors have provided the reply to the Referee 2's comment, and are invited to do the same for the the Referee 1's comment as well.

As an Editor, I have to take a quite "independent view", but I am quite interested in this subject, so decided to write a short comment, and thus to contribute to the discussion.

I would like to mention that - indeed - the paper would benefit if the difference between the presented approach and the more traditionally used sensitivity analysis (SA) and uncertainty analysis (UA) methods is explained clearer. If I understand it correctly, the presented method consists of the following: a) randomly sample parameters (using LHS) and run the model; b) using generated data build a surrogate model (RT) of the response surface (for each perf.crit.); c) estimate "strength" of relationship by looking

at the "percentage contribution of each model parameter in explaining the variability in a certain performance criterion". Both reviewers mention that this can be seen as a variation of SA (and even Monte Carlo based UA) - albeit, in my opinion, with an interesting twist of using a surrogate model and the way "strength" is estimated. However the idea of "propagating" variation (sampling) in parameters through a model, and estimating how much does it influence the output (or performance) can be seen by many readers as similar to SA and UA. So, again, the difference could be perhaps presented more convincingly.

A comment on RT: it is known that it is not the most accurate machine learning model: in its canonical form, its output in each leave is a zero-order regression model (i.e. a constant), whereas e.g. M5 model tree (Quinlan 1986) generates the 1-order (linear) regression model (unless R code of RT does this differently). (However RT has an advantage that is is simple.) Of course there are also many other methods like ANN. Would be useful to compare if (how) results using RT differ from the results if another type of the surrogate (approximating) model is used.

I hope this comment can be also taken on board.

---

## Referee Comment (RC3) · R. Arsenault (Referee) · 1 Apr 2017

This study uses regression trees in a bi-directional framework to estimate the importance of a model parameter in a set of objective functions, as well as the relative importance of each parameter to a given performance measure. The authors conclude that this method permits identifying model parameters with respect to certain performance measures, and I agree with their assessment. I found this paper to be interesting and generally well written. I have a few important comments which I would like to see addressed before considering publication in HESS. I recommend a major revision.

First, the author's RT based method should ideally be compared to other proven techniques to estimate parameter importance, such as Sobol' sensitivity analysis (or any global sensitivity analysis). If not integrated into the work directly, differences in expected outcomes should be addressed in the literature review. I recognize that the bi-directional aspect of this work is novel but any advantages of this method should be compared to a proven baseline.

Page 4, lines 4-6: This sentence is very confusing, please rewrite differently.

Page 4 lines 7-8: This means that the model parameter has no relevant impact on other performance measures. Perhaps give a clear example of how this can be achieved in the case of a hydrological model with highly interacting parameter sets. My previous work in parameter identifiability suggests that a large part of the relative importance of a parameter on a performance measure comes from its interactions with other parameters.

Page 7, lines 4-5: By using 2000 samples with hypercube sampling, are the authors not effectively working in spaces where parameter combinations might not make physical sense? Usually the model parameters, during calibration, will self-regulate to attain sensible parameter values. With a LHS approach, perhaps some combinations are tested here which are out of the bounds that the model can work with appropriately. More information regarding this aspect would be interesting.

Furthermore, the parameters do not seem to be normalized in their ranges, therefore allowing some parameters more leverage over the performance measures. If I interpreted this correctly, then some of the results would be trivial since the larger boundaries will naturally have more effect on the performance measure and thus the parameter will be more "important". The use of a LHS methodology in an uneven search space will bias the results (as an extreme example, if ESCO bounds were set between 0.995 and 1.005, then the parameter would definitely not be considered important). The choice of boundaries, then, induces a methodological bias in the results. I am not sure how to solve this problem, perhaps by performing multiple calibrations and taking the envelope of the parameter sets, but this also has its drawbacks.

Also, the parameters seem to be evaluated on the entire time series. In a snowmeltdominated catchment, the parameters are highly time-variant. How could this affect the method's robustness?

I think Figure 1 can be omitted completely without any loss of information in the paper. It is fairly well described in the text.

Page 12, lines ~20-25: I have the feeling that some of these strong connections are trivial. If I had had to guess in advance, I would have guessed that Evapotranspiration (ESCO) is probably strongly linked to bias (KGE_beta), and that mid flows and lower were also affected by baseflow recessions and to some extent evaporation due to the relative scale of a fixed evaporation rate on total available volumes. Once interactions are important, then the method seems to "get lost" in a sense, as there is no clear path to identifiability (as demonstrated in the discussion). I think sensitivity analyses would provide the same information while also informing on the different order sensitivities.

---

## Editor Comment (EC2) · D. Solomatine (Editor) · 12 May 2017

The discussion was intersting and obviously there have been points raised which will lead to improvements in teh manuscript. They are clearly presentd by referees, and it seem the auhotrs agree. Some clarifications are needed in presenting the suggestd approach and to make distinction from UA and SA clearer. I have mention a number of points in the "intermediate comment" as well. I wish the authors success in revising th manuscript.

---

## Author Comment (AC2) · 12 May 2017

Comment: This study presents an approach to quantify the strength of the bijective relationship between model parameters to performance measures. The proposed method explores the model parameter-performance space using regression trees with the goal to detect performance measures that can uniquely identify a parameter. The regression trees are first developed by casting model parameter as explanatory variables and performance measure as prediction variables, and then by exchanging the explanatory and prediction variables. These trees are developed for two catchments in Germany. The main idea presented in the study is interesting and results contribute valuable insights towards model diagnostics.

Reply: We thank the reviewer for summarising our methodology and for emphasising

the value of our study for the hydrological community.

C: However, there are a few issues that should be addressed. First, the introduction requires revision so as to remove repetition of ideas (see for example, lines 11 and 32 on Page 2), and to provide more background.

R: In the revised version of the manuscript, we will rework the introduction, as also raised by comments from the other referees and the Editor. In this way, we will reduce repetitions in the introduction. As also raised by the other reviewers, we will add a discussion of how our study is related to sensitivity analysis to improve the overall presentation of our study.

C: The need for multiple performance measures to identify unique aspects of the hydrograph is well motivated, but why this has remained a challenge is not discussed.

R: We certainly agree that the use of multiple performance criteria was already emphasised. It is known that each performance criterion is related to different aspects of the hydrograph. However, it is still unclear a) which performance criteria are appropriate for parameter value selection and b) which performance criteria are related to which (type of) parameters. We will address this issue also by including additional references to this point when revising the introduction and provide a more focused introduction in the revised version of the manuscript.

C: Some well-known issues are parameter interaction, limited information content in hydrologic time series data that allows identification of only a handful of parameters, and uncertainties in input as well as streamflow data (Beven, 2011).

R: Certainly, parameter interaction and limited information content complicate the interpretation of model results. These points also complicate the understanding which and how model parameters control the hydrological behaviour in models. We see our approach as a contribution to a more detailed use of available information with the aim to achieve a more precise identification of all relevant model parameters. We have

shown how the use of different performance criteria help in identifying model parameters. Thus, based on our approach the parameter identification can be improved systematically. The connective strength shows how precisely a model parameter can be identified based on the existing information content. Since we selected a two-step approach by using at first the performance criteria and secondly the model parameters as explaining variables, we can provide some statements to the aspect of parameter interaction. In RTpar with the model parameters as explaining variables, each model parameter is individually assessed. There is an indirect impact since the model parameter values are different for each model simulation. Thus, by comparing RTpar between the different model parameters runs and thus different model parameter sets, we can see whether the performance criteria which are influenced by changes in this parameter are the same. Moreover, in comparing the ten RTperf applications, we can see whether the same model parameters affect different performance criteria. This is also an insight for parameter interaction. Thus, our study contributes to better parameter identification also under consideration of parameter interactions. We will consider this point while revising the mauscript.

C: Complicating this further is the time varying nature of parameter sensitivity (Herman et al. 2013).

R: We agree that we have to add the current state-of-the-art in time-varying sensitivity analyses to the introduction. Moreover, we will relate our approach in the discussion to the recent approaches in sensitivity analyses. Since we have presented studies on sensitivity analyses during the last years (Guse et al., 2014, 2016a,b), in the revised version of the manuscript we will refer to these studies by also considering other major recent studies on this topic such as Herman et al. (2013a,b) or Van Werkhoven et al. (2008).

C: Another important issue is the comparison of the proposed method to already existing sensitivity analysis methods, which also attempt to identify relationship between model performance and parameters. In my understanding, it is the partitioning of the

model performance space by parameter values that is unique about regression trees, but the order of importance of parameters should ideally be the same as that derived using sensitivity analysis methods.

R: We agree with the reviewer that in general similar results in parameter ranking are expected in sensitivity analysis. This can be proved by comparing this manuscript with our recent work (Guse et al., 2016a,b) on sensitivity analyses on model results (and not on performance criteria). The parameter ranking could be different when using different performance criteria. This was shown in our study, since we focused more on the relationship between performance criteria and model parameters. We examine which performance criteria are appropriate to identify at best the parameter values. The novelty is that we assessed this relationship bijectively from both sides. At first, as usual we used performance criteria as target variable (such as typically in sensitivity analysis) and secondly we used model parameters as target variable. This cannot be realized in a sensitivity analysis and is a clear benefit of our approach compared to sensitivity analysis. However, as described in the previous comment, we will relate our approach to sensitivity analyses in the revised version of the manuscript and discuss the advantage of our approach in the discussion chapter.

Specific Comments:

C1. Method description: Further information on implementation of regression trees is warranted. For example, the metric: 'percentage contribution of each explaining variable' is used throughout the manuscript without an explanation to how it is actually estimated by regression trees. It is expected that any method will have some error or uncertainty associated with its results, so what levels of 'percentage contribution' are significant? If any cross validation analysis during tree construction was used, it should be explained.

R1: We will improve the description of percentage contribution in the revised version of the manuscript. In this context, we are not aware of a threshold value stating that

a percentage contribution is relevant above this value. Within the construction of the regression tree, a cross-validation analysis was automatically realised in the r-package rpart (Therneau and Atkinson, 2010). Hereby, it was investigated how an additional split leads to a better explanation.

C2. Methodological choices: It is mentioned in the manuscript that it is likely some performance measures are correlated (Lines 6-9, Page 7), these correlations are also presented in Figure 2. However, all performance measures are used for generating trees for RTpar. Could this potentially be the reason behind poor connective strength between performance measures and parameters for high flows? It should be discussed whether the regression tree algorithm can deal with correlated input. If not, correlated performance measures should ideally be reduced to an uncorrelated set. In fact, the same holds true for the use of model parameters as independent variables in RTperf, the presence of parameter interaction will affect the results to some extent.

R2: The selection of the performance criteria impacts the results of RTpar. Using similar performance criteria, the percent contribution for single performance criterion may decrease. However, as the comparison between both catchments shows, there are large differences in the results of the two catchments. The high similarities in five performance criteria in the Treene catchment are not observed in the Saale catchment. The idea is to cover different aspects of hydrological behaviour by using these ten performance criteria which consider different aspects of hydrological behaviour. Depending on catchment characteristics and the way the model can produce the hydrological behaviour in these catchments, the impact of performance criteria might be different. Thus, it is not fully clear in advance which performance criterion is best suited to represent a certain model parameter. Due to that, we are convinced that it is required to use all these performance criteria. In addition, we tried to cover with our selection the most common performance criteria as well as the different aspects of hydrographs. However, the issue of correlated performance criteria needs to be certainly considered in the interpretation of connective strength. Concerning model parameters, we do not

agree to remove model parameters because of parameter interactions in RTperf. The idea is to detect which model parameters impact a certain performance measure. In this context, it would not be helpful to remove interacting model parameters. We are not aware of a similar approach in model calibration. In the case of two relevant, but correlated model parameters, we have nevertheless to identify appropriate values for both model parameters and not only for the most relevant one.

C3. Background data: The time period of analysis, values of catchment average precipitation, temperature, etc. should be provided. An appendix with some details on the model structure and implementation of SWAT can be considered to make the study independent of prior applications of the model to these catchments. As the main focus is model diagnostics, it is essential that readers are aware of the model structure and the details of its implementation.

R3: We agree that more information of the SWAT model would be helpful to understand our model diagnostic approach. Thus, we will provide more detailed information of the SWAT model in the revised version of the manuscript.

C4a. Issue of CN2 (Line 16, Page 11): It is surprising that no appropriate performance criteria is found to relate to CN2, which is generally a sensitive parameter in SWAT. One reason can be the low variation assigned to it (only within +/- 10 of base value, see Table 1). On the other hand, some other parameters are allowed to vary within much larger ranges (GW_DELAYfsh between 1-50, RCHRGssh between 0.2-0.8, etc.). It is later found that these parameter display high connectivity to performance measures.

R4a: We have selected parameter ranges based on former studies (Guse et al., 2014, 2016a, Pfannerstill et al., 2014, 2015) and we think that they are well justified. Even for other model parameters such as RCHRGssh which can range from 0 to 1, we have reduced the parameter ranges to minimize the number of inappropriate model simulations. The parameter ranges were selected so that the processes are adequately represented and unrealistic parameter combinations (values) are intended to be avoided.

An increase of parameter ranges can always result in a higher risk of unrealistic high or low relevances of a certain hydrological component. In our experience, an increase in the parameter range can increase the impact of a model parameter in investigating its influence on a performance criterion. However, we are not aware of an example of a SWAT model in which a parameter with a low relevance becomes strongly relevant after increasing a well-justified parameter range. In the case of CN2, we have to add that the initial parameters were already carefully checked. Thus, a value for a certain land use type of 50 means that the CN2 was varied in these HRUs from 40 to 60 which is assumed to be sufficient to maintain the landscape heterogeneity. Moreover, we like to highlight that surface runoff is only of low relevance in the Treene catchment and of medium relevance in the Saale catchment. A higher relevance of CN2 would be expected for catchments with higher relevance of surface runoff. Thus, the selected ranges are seen as representative to reproduce the process accurately and to avoid unrealistic high or low contributions of surface runoff.

C4b: Please also mention the units of parameters in Table 1.

R4b: Units will be added.

C5. Threshold of performance: Figure 1 shows that negative NSE and KGE values were also allowed in the tree construction. The issue of using parameter sets related to highly degraded performance has been raised and addressed by earlier studies (Kelleher et al. 2013). Should a threshold of performance be fixed and only those parameter sets that perform above it considered for further analysis?

R5: We are aware that the performance of model runs is different and that a reduction of a certain performance criteria shows a lower quality of the model run. However, we are not aware of a consistent approach to identify thresholds for "good" and "poor" model runs. All selections of thresholds are somehow arbitrary. Thus, we prefer to use the whole data set of 2000 model simulations. However, we can add to the manuscript that the values of the performance criteria can have a high range and that the ranges

are shown in the correlation plots.

C6. Convergence of results with number of LHS samples: 2000 parameter sets are used in the analysis but no discussion on the stability of results w.r.t number of LHS samples is provided. One way to test this is to look at the agreement between current results with those from a subset of 500, and 1000 sets. Typically, the number of sets after which little fluctuation in results is seen is used.

R6: We see 2000 parameter sets as a good number to represent the parameter space accurately for our purpose. According to our experiences with the SWAT model and the Latin-Hypercube sampling a reduction to 500 and 1000 does not lead to a good coverage of the parameter space. In this case, the information content might be too low for a realistic construction of regression trees, since the number of model runs within a subset reduces from node to node and in the case of 500 models finally the number of model runs is too low in a subset to provide reasonable results.

C7. Equation 3, Page 7: Please elaborate how the RSR calculation is implemented. Say there are only 10 flow values for 0-5 percentile range for observed flow but 100 such values are present for simulated flow, how is RSR then calculated?

R7: Both, observed and simulated discharge time series are available for the same modeling period, i.e. both have the same length. In constructing the FDCs, both time series are separately considered. However, the number of days for 5% of the total time series is identical. Thus, we have the same number of flow values for example in the 0-5 percentile range. Based on this, the equation for the RSR can be applied.

Technical Corrections

C1. Line 1, Page 1: Consider replacing 'parameters are used to adapt the model to the conditions of the catchment' with 'parameters are used to represent the time-invarying characteristics of the catchments'.

R1: Changed.
C2. Line 1, Page 2: Consider replacing 'In models' with 'In rainfall runoff models'.

R2: Changed.

C3. Lines 7-9, Page 2: It is now generally accepted that parameters may or may not be identifiable (Beven, 2011).

R3: We are aware of studies on parameter identifiability. However, we still think that it might be worth to investigate the parameter identifiability using different performance criteria and to understand why a certain model parameter cannot be identified and whether this is related to the selection of the performance criteria.

C4. Line 21, Page 4: Explain 'high drainage activities'.

R4: High drainage activities mean that a high percentage of agricultural areas is covered by drainages. We will describe in the revised version of the manuscript that large parts of agricultural areas are drained.

C5. Line 28, Page 4: Replace 'temporally' with 'temporal'.

R5: changed

C6. Line 22, Page 11: Remove 'extremely'.

R6: changed

C7. The text size in Figure 3 (lower panel) should be increased for visibility.

R7: changed

References:

Guse, B.; Reusser, D. E.; Fohrer, N. (2014): How to improve the representation of hydrological processes in SWAT for a lowland catchment - Temporal analysis of parameter sensitivity and model performance, Hydrol. Process., 28: 2651–2670. doi: 10.1002/hyp.977.

Guse, B.; Pfannerstill, M.; Strauch, M.; Reusser, D.; Lüdtke, S.; Volk, M.; Gupta, H.; Fohrer, N. (2016a): On characterizing the temporal dominance patterns of model parameters and processes, Hydrol. Process., 30(13), 2255-2270, doi:10.1002/hyp.10764.

Guse, B.; Pfannerstill, M.; Gafurov, A.; Fohrer, N.; Gupta, H. (2016b): Demasking the integrated information of discharge: Advancing sensitivity analysis to consider different hydrological components and their rates of change, Water Resour. Res., 52, 8724-8743, doi:10.1002/2016WR018894.

Herman, J.D.; Kollat, J.B.; Reed, P.M.; Wagener, T. (2013a): From maps to movies: high resolution time-varying sensitivity analysis for spatially distributed watershed models. Hydrology and Earth System Sciences, 17, 5109–5125.

Herman, J.D.; Reed, P.M.; Wagener, T. (2013b): Time-varying sensitivity analysis clarifies the effects of watershed model formulation on model behavior. Water Resources Research, 49, doi:10.1002/wrcr.20124.

Pfannerstill, M.; Guse, B.; Fohrer, N. (2014): Smart low flow signature metrics for an improved overall performance evaluation of hydrological models, J. Hydrol, 510, 447-458, doi:10.1016/j.jhydrol.2013.12.044.

Pfannerstill, M.; Guse, B.; Reusser, D.; Fohrer, N. (2015): Process verification of a hydrological model using a temporal parameter sensitivity analysis, Hydrol. Earth Syst. Sci., 19, 4365-4376, doi:10.5194/hess-19-4365-2015.

Therneau, T.M. and Atkinson, B. (2010): Rpart: Recursive partitioning. R package, http://CRAN.R-project.org/package=rpart.

van Werkhoven, K.; Wagener, T.; Reed, P.; Tang, Y. (2008): Characterization of watershed model behavior across a hydroclimatic gradient. Water Resources Research 44: W01429. doi: 10.1029/2007WR006271

---

## Author Comment (AC3) · 12 May 2017

Comment: This study uses regression trees in a bi-directional framework to estimate the importance of a model parameter in a set of objective functions, as well as the relative importance of each parameter to a given performance measure. The authors conclude that this method permits identifying model parameters with respect to certain performance measures, and I agree with their assessment. I found this paper to be interesting and generally well written. I have a few important comments which I would like to see addressed before considering publication in HESS. I recommend a major revision.

Reply: We thank Richard Arsenault for this very positive statement to our manuscript and for encouraging us to revise the manuscript.

[Figure]

C: First, the author's RT based method should ideally be compared to other proven techniques to estimate parameter importance, such as Sobol' sensitivity analysis (or any global sensitivity analysis). If not integrated into the work directly, differences in expected outcomes should be addressed in the literature review. I recognize that the bi-directional aspect of this work is novel but any advantages of this method should be compared to a proven baseline.

R: We agree that the manuscript would benefit by introducing a discussion on sensitivity analyses also by considering the comments from the other referees and the Editor. Since we have presented studies on sensitivity analyses during the last years (Guse et al., 2014, Pfannerstill et al., 2015, Guse et al. 2016a,b), we will refer to these studies and consider other major recent studies on this topic such as Herman et al. (2013a,b) or Van Werkhoven et al. (2008, 2009).

In the current version of the revised manuscript, the added passage at the end of the discussion to this issue reads as stated below. This text is in an intermediate state and may change during the revision.

"The relevance of model parameters can also be investigated using sensitivity analyses. By comparing the parameter relevances with former studies on temporally resolved parameter sensitivity analyses, it becomes apparent that the overall ranking is similar (Guse et al., 2014, 2016b). While we used here ten different performance criteria in regression trees, the parameter sensitivities were separately derived for the five segments of the FDC. In both cases, the differences in parameter relevances between different hydrological conditions were shown and are consistent.

As a further development, the bijective analysis of the relationship between model parameters and performance criteria was introduced here. The interpretation of this relationship from sites is an advantage compared to classical analysis of the impact of model parameters on performance criteria such as realised in sensitivity analyses (van Werkhoven et al., 2008; Herman et al., 2013a,b). With our approach, it is possible to

investigate which performance criteria are appropriate for a certain model parameters.

A core advantage of RT is that subsets of the simulation runs are constructed in a structured way. By subdividing the simulation set based on the major influencing variables at each branch, two distinct subsets occur which differs in the values of model parameters as well as of the performance criteria. As shown in the example of a RT, good model simulations are separated from poor simulations. And also the parameter values within each subset are different. With this subset construction, it can be detected whether a model parameter has the highest explanatory power in a certain branch."

C: Page 4, lines 4-6: This sentence is very confusing, please rewrite differently.

R: We are aware that we have to explain the RTpar method in a clearer way. In the revised version of the manuscript, this sentence will be rewritten.

C: Page 4 lines 7-8: This means that the model parameter has no relevant impact on other performance measures. Perhaps give a clear example of how this can be achieved in the case of a hydrological model with highly interacting parameter sets. My previous work in parameter identifiability suggests that a large part of the relative importance of a parameter on a performance measure comes from its interactions with other parameters.

R: One aspect of introducing the connective strength is to show whether a model parameter can be clearly identified by one specific performance criterion. This means that this performance criterion is precisely related to the process which is related to this parameter. One example is the strong bijective relationship between the baseflow recession coefficient ALPHA_BFssh and the RSR for the very low segment. Despite of parameter interaction, it becomes apparent that this parameter is best identified by the RSR for the very low segment which makes sense, since ALPHA_BFssh is the major low flow parameter. Moreover, we like to mention another point: Our core aim is to improve parameter identification. Thus, in the best we want to consider a model parameter in isolation. Thus, a model parameter which is always insensitive or only due to parameter interaction is in both cases a parameter with a low parameter identifiability.

C: Page 7, lines 4-5: By using 2000 samples with hypercube sampling, are the authors not effectively working in spaces where parameter combinations might not make physical sense? Usually the model parameters, during calibration, will self-regulate to attain sensible parameter values. With a LHS approach, perhaps some combinations are tested here which are out of the bounds that the model can work with appropriately. More information regarding this aspect would be interesting.

R: We agree with the reviewer that within a parameter sampling, different qualities of parameter combinations are considered. Certainly, some of them make more sense than others. This aspect is considered at first by constraining the parameter ranges according to our experiences to reduce unrealistic parameter combinations, which lead to non-physically process description. We will more clearly state in the revised version of the manuscript that we constrain the parameter space to reduce unrealistic parameter combinations. Second, by calculating the performance criteria, we detect the quality of the model runs. However, since the selection of thresholds on performance criteria is somehow arbitrary, we think that it is a more consistent approach to do not restrict the data set in a second step due to the performance criteria. Otherwise we would have to justify ten thresholds (for each performance criterion). In this context, we like to emphasise that the problem of analysing parameter sensitivities with model simulations of different performance appears also in sensitivity analyses. In particular, studies with multiple performance criteria show that it is very difficult to subdivide a data set into "good" and "poor" performing model runs.

C: Furthermore, the parameters do not seem to be normalized in their ranges, therefore allowing some parameters more leverage over the performance measures. If I interpreted this correctly, then some of the results would be trivial since the larger boundaries will naturally have more effect on the performance measure and thus the parameter will be more "important". The use of a LHS methodology in an uneven

search space will bias the results (as an extreme example, if ESCO bounds were set between 0.995 and 1.005, then the parameter would definitely not be considered important). The choice of boundaries, then, induces a methodological bias in the results. I am not sure how to solve this problem, perhaps by performing multiple calibrations and taking the envelope of the parameter sets, but this also has its drawbacks.

R: We are aware of this problem. As it is also typical in sensitivity analyses, the selection of the lower and upper bounds of the parameter values is critical. However, we have selected the parameter ranges based on several former studies with the SWAT model and think that we have selected reasonable ranges (Guse et al., 2014, 2016a, Pfannerstill et al., 2014, 2015). Our experience with increasing ranges (within reasonable areas) is that the general pattern of the results is not largely affected.

C: Also, the parameters seem to be evaluated on the entire time series. In a snowmelt-dominated catchment, the parameters are highly time-variant. How could this affect the method's robustness?

R: Certainly, the results would be different if we would apply it only to a sub-period or only to winter month. However, we think that this issue is still a big challenge in hydrological modelling. For example, we are not aware of a consistent approach how to identify or calibrate snow parameters using daily resolution and a reasonable sub-period. It is still challenging to consider sub-periods for model calibration. We agree that this issue is relevant, but in our opinion out of the scope for this study. However, we could mention this point at the end of the discussion as an outlook for future research.

C: I think Figure 1 can be omitted completely without any loss of information in the paper. It is fairly well described in the text.

R: Based on the positive feedback during the presentation of this work at the General Assembly of the European Geosciences Union (EGU) in April 2017 in Vienna, we intend to keep this figure in the revised manuscript.

C: Page 12, lines _20-25: I have the feeling that some of these strong connections are trivial. If I had had to guess in advance, I would have guessed that Evapotranspiration (ESCO) is probably strongly linked to bias (KGE_beta), and that mid flows and lower were also affected by baseflow recessions and to some extent evaporation due to the relative scale of a fixed evaporation rate on total available volumes. Once interactions are important, then the method seems to "get lost" in a sense, as there is no clear path to identifiability (as demonstrated in the discussion). I think sensitivity analyses would provide the same information while also informing on the different order sensitivities.

R: We agree that some results could be expected. To give an idea of this issue, we have selected two contrasting catchments. Our results clearly show that there are lots of differences even in these two catchments. Moreover, we agree that some relationships such as between ESCO and KGE_beta are not surprising. On the other side, other relationships such as between the curve number and the RSR for high flows do not appear to be strong, even if this could be expected as well. Thus, we think that it worth to check the relationships whether they are really as relevant as expected. Most importantly, a sensitivity analysis would only provide one-directional results, while our approach is focused on the bijective relationship between model parameters and performance criteria.

References:

Guse, B.; Reusser, D. E.; Fohrer, N. (2014): How to improve the representation of hydrological processes in SWAT for a lowland catchment - Temporal analysis of parameter sensitivity and model performance, Hydrol. Process., 28: 2651–2670. doi: 10.1002/hyp.977.

Guse, B.; Pfannerstill, M.; Strauch, M.; Reusser, D.; Lüdtke, S.; Volk, M.; Gupta, H.; Fohrer, N. (2016a): On characterizing the temporal dominance patterns of model parameters and processes, Hydrol. Process., 30(13), 2255-2270, doi:10.1002/hyp.10764.

[Figure]

Guse, B.; Pfannerstill, M.; Gafurov, A.; Fohrer, N.; Gupta, H. (2016b): Demasking the integrated information of discharge: Advancing sensitivity analysis to consider different hydrological components and their rates of change, Water Resour. Res., 52, 8724-8743, doi:10.1002/2016WR018894.

Herman, J.D.; Kollat, J.B.; Reed, P.M.; Wagener, T. (2013a): From maps to movies: high resolution time-varying sensitivity analysis for spatially distributed watershed models. Hydrology and Earth System Sciences, 17, 5109–5125.

Herman, J.D.; Reed, P.M.; Wagener, T. (2013b): Time-varying sensitivity analysis clarifies the effects of watershed model formulation on model behavior. Water Resources Research, 49, doi:10.1002/wrcr.20124.

Pfannerstill, M.; Guse, B.; Fohrer, N. (2014): Smart low flow signature metrics for an improved overall performance evaluation of hydrological models, J. Hydrol, 510, 447-458, doi:10.1016/j.jhydrol.2013.12.044.

Pfannerstill, M.; Guse, B.; Reusser, D.; Fohrer, N. (2015): Process verification of a hydrological model using a temporal parameter sensitivity analysis, Hydrol. Earth Syst. Sci., 19, 4365-4376, doi:10.5194/hess-19-4365-2015.

van Werkhoven, K.; Wagener, T.; Reed, P.; Tang, Y. (2008): Characterization of watershed model behavior across a hydroclimatic gradient. Water Resources Research 44: W01429. doi: 10.1029/2007WR006271

van Werkhoven, K.; Wagener, T.; Reed, P.; Tang, Y. (2009): Sensitivity-guided reduction of parametric dimensionality for multi-objective calibration of watershed models. Advances in Water Resources 32(8): 1154–1169.

---

## Author Comment (AC4) · 12 May 2017

Comment: The authors have provided the reply to the Referee 2's comment, and are invited to do the same for the Referee 1's comment as well.

Reply: By replying directly to referee#2, we intend to clarify misunderstandings. In the meantime we replied also to the first and the third referee.

C: As an Editor, I have to take a quite "independent view", but I am quite interested in this subject, so decided to write a short comment, and thus to contribute to the discussion.

R: We thank the Editor to take part at this stage of the discussion and clarify these aspects as emphasised below.

[Figure]

C: I would like to mention that - indeed - the paper would benefit if the difference between the presented approach and the more traditionally used sensitivity analysis (SA) and uncertainty analysis (UA) methods is explained clearer.

R: We agree with the Editor and also with the referees that a clearer differentiation of our approach in comparison to a sensitivity or uncertainty analysis would improve the manuscript. This might be a good point for the discussion. In the revised version of the manuscript, we will add this part. Moreover we will relate the core idea of our manuscript to sensitivity analyses in the introduction by representing the state-of-the-art in relation to the objectives of our work.

C: If I understand it correctly, the presented method consists of the following: a) randomly sample parameters (using LHS) and run the model; b) using generated data build a surrogate model (RT) of the response surface (for each perf.crit.); c) estimate "strength" of relationship by looking at the "percentage contribution of each model parameter in explaining the variability in a certain performance criterion".

R: Overall, we agree with this short description of our approach and thank the Editor for this short summary. To a) We like to add that the calculation of ten performance measures for each model run is part of the point. To b) The regression trees are not only built for each performance measure (RTperf, see Fig. 4) but also in a second step for each model parameter using the performance measures as explaining variables (RTpar, see Fig. 5). Thus, we looked at the relationship between model parameters and performance measures from both sides by interpreting the bijective relationship (connective strength). To c) The (connective) strength of the relationship between model parameters and performance measures is described by considering the percentage contribution as described above by the Editor and also vice-versa by estimating the percentage contribution of how a certain performance measure react on changes in the parameter values (see Fig. 6). We will improve this description in the revised version of the manuscript.

C: Both reviewers mention that this can be seen as a variation of SA (and even Monte Carlo based UA) - albeit, in my opinion, with an interesting twist of using a surrogate model and the way "strength" is estimated. However the idea of "propagating" variation (sampling) in parameters through a model, and estimating how much does it influence the output (or performance) can be seen by many readers as similar to SA and UA. So, again, the difference could be perhaps presented more convincingly.

R: We agree that our manuscript would benefit from relating our method to sensitivity and uncertainty analysis and will incorporate the suggested differentiation in the revised version of the article. We like to highlight that a major part of our study is the analysis of the bijective relationship between model parameters and performance criteria. Thus, we do not only investigate how variations are propagated in the model up to the output but also by looking which outputs (i.e. performance criteria) are impacted by a certain model parameter.

C: A comment on RT: it is known that it is not the most accurate machine learning model: in its canonical form, its output in each leave is a zero-order regression model (i.e. a constant), whereas e.g. M5 model tree (Quinlan 1986) generates the 1-order (linear) regression model (unless R code of RT does this differently). (However RT has an advantage that is is simple.) Of course there are also many other methods like ANN. Would be useful to compare if (how) results using RT differ from the results if another type of the surrogate (approximating) model is used.

R: We agree that also other approaches than RTs are possible. However as raised in this comment, RT is a simple approach which can be relatively easily applied and understood. Since the suggested approach is not common we do not want to make it even more complicated. Our main focus was to emphasise the bijective strength between model parameters and performance measures. Thus, we think that the selection of RT is justified. Moreover, we do not expect that the results will significantly differ through the application of one of the mentioned approaches. Our results of parameter relevance for different hydrological conditions coincide with the results of sensitivity

analysis in former studies. Thus, we want to keep the current focus of the manuscript.

C: I hope this comment can be also taken on board.

R: We will certainly also consider these comments while revising the manuscript.

————————————————————

---

## Author Response (AR2)

Comments from the Editor:

Dear authors

As you can see from reviews, two of them are quite positive (one suggest minor updates), but the one of Referee 3 (report 2) is not - he/she recommends rejection.

I also think this paper presents an interesting approach worth attention of scientific world. So in this respect I am with the two reviewers out of three. Here there is some disagreement, but it is just normal in the world of researchers.

At the same time - I would still strongly suggest to take on board (some of) the comments of the referee who suggests rejection. He/she also has a point - and this opinion can be shared by more people (please also see comments, including mine, at earlier stages - on clearly stating the difference with the traditional approaches.) In this current version, these differences are much better explained already, but still I would ask to address the raised points again. I think it is not too difficult to do.

Good luck.

We thank the Editor for this positive feedback.

Please find our replies to the comments from the reviewers below.

We like to highlight that as main modification we revised the introduction of the manuscript according to the comments from reviewer#1. Moreover, a map of the catchment sites was added as recommended by reviewer#2.

Our text is coloured as follows:

- Reviewer' comments
- Answer
- „Text directly taken from the revised version of the manuscript"
- **„Modified parts of the text directly taken from the revised version of the manuscript"**

First reviewer:

In the revised version of the manuscript, the authors have provided clarifications for study area and methodological choices. In addition, introduction has been edited to account for comments by the referees and editor. The manuscript has overall improved, but the following minor issues still remain unaddressed:

1. Unclear reasoning in the introduction: Some claims in the introduction are not substantiated by past literature. For instance:

a. Lines 10-13, Page 2 ('In parameter identification ………. Identification'): It is not true that parameter identification implicitly assumes that model parameters are precisely identified by selected performance criteria. Most sensitivity analysis studies have already shown that this is not true, and that parameters can affect performance in interaction with other parameters within the model. In fact in some cases, the interaction effects dominate the sensitivity indices. In fact, the assumption that an accurate parameter identification can be done is questionable given the interaction effects and uncertainties.

We agree that this part of the introduction needs to be improved. Thus, we revised the second paragraph of the introduction carefully. During the revision, the marked sentence was removed.

b. Line 25-28, Page 3 ('To our …. Task.'): There is a misinterpretation of sensitivity analysis here. If one applies sensitivity analysis with multiple performance measures, a matrix similar to Figure 4 can be obtained, where gray scales can be used to plot the first or higher order contributions of parameters to variation in each measure. A comparison across performance measures can then highlight which parameters uniquely identify which parameter measures and vice-versa. In fact, applying sensitivity analysis in this manner will give more information than the framework proposed in this study as they can also highlight higher order interaction effects. I think this study is not an advancement on sensitivity analysis but an alternative. Both methods can be used to attain the overall goals of the analysis. The only advantage of using CART over SA is that CART identifies the values of parameters that lead to good performance while SA does not. If used only for parameter/performance ranking, both are equally acceptable methods.

We certainly agree that our approach is an alternative to sensitivity analyses. And we also agree that the impact of parameter interactions can be better assessed in a sensitivity analysis method which also considers parameter interactions.
However, we think that the bijective approach in our study is new. The classical approach of identifying the best parameters for multiple performance criteria was enhanced by looking from the side of the model parameters. In our approach we used the entire set of model simulations in RT to identify the most appropriate performance criteria for a given model parameter. Here, to construct RTs only the set of performance criteria and the selected model parameter is used. Thus, all other model parameters are not directly included. They were only indirectly considered since the variation of their values also impacts the modelled discharge time series and thus the performance criteria. Nevertheless since only the selected model parameter is included in RT, the most appropriate performance criteria for handling this parameter can be precisely identified. And moreover, it is can identified whether one of the selected performance criteria is suitable to identify an adequate value for the selected model parameter.
We agree that in a sensitivity analysis, it is possible (and helpful) to compare the sensitivity values of model parameters for multiple performance criteria. However, with a sensitivity analysis it is not possible to detect the most appropriate performance criteria for a given model parameters (in particular for model parameters of low relevance). The performance criteria for which the highest sensitivity value was computed for a given model parameters is

not undoubtedly the most appropriate performance criteria. It could be that this performance criterion is not strongly impacted by variations in model parameters.

When comparing the results of both RT approaches (Figs. 5 and 6 in the revised version of the manuscript), it becomes apparent that the performance criterion with the highest percent contribution for a given model parameter in Fig. 5 is in several cases not identical with the performance criterion with highest percent contribution in Fig. 6. Thus, the analysis from the side of the model parameters provides different results and new knowledge about the interrelationship between model parameters and performance criteria. While the interpretation of the results in Fig. 5 is possible in a similar way with a sensitivity analysis, the Fig. 6 provides results which cannot be derived in a sensitivity analysis. This aspect is added as a new paragraph in the discussion (see P. 15 L. 13-16).

2. Choice of 2000 parameter sets: this choice is still poorly defended. If one has 12 parameters and each parameter range is divided into 10 equally spaced values, one still needs 10^12 values to represent the parameter space. 2000/10^12 is a very sparse set and cannot be claimed to represent the parameter space accurately. Therefore, a convergence analysis is the only realistic way to defend this choice. It is hard to say whether 2000 is far better than 1000. It is unclear why the results from a randomly chosen subset of 1000 (or 1500, or 1800) cannot be checked against the results presented in the main analysis, as a proof of concept.

In general we agree that 2000 model simulations are not enough to identify parameter values such as it is realized in calibration studies. However, our intention was to investigate the relationship between model parameters and performance criteria. Thus, our study is as discussed in the previous comments more related to a sensitivity analysis than to a model calibration. In a sensitivity analysis, the number of required model simulations can be lower such as in our experiences with the FAST model approach. Thus, to investigate the relationship between model parameters and performance criteria, we think that this number of model simulations is appropriate.

Nevertheless, as also discussed in the first review round, we checked how the patterns of our results are impact by a different number of simulations (500 and 3000). While the results for 500 model simulations are largely different from ours, the results for 3000 are similar. Thus, we think that our statements are supported by 2000 model simulations.

In this context, we like to mention that we had previous experiences with the SWAT model in both catchments. A better knowledge of how the model parameters react helps to select reasonable model parameters and reasonable parameter ranges. Based on this, the amount of unrealistic parameter combinations was already reduced. Thus, we think that the number of 2000 model simulations is also justified since more realistic parameter combinations are included as it would be in a first application in a new catchment. Certainly, the number of required model simulations might be higher if using a new model or an unknown catchment.

According to this comment, we extend the paragraph on model simulations in the revised manuscript for a better justification of our procedure:

The paragraph reads now as follows (P.5, L. 31 to P. 6, L.10):

**Based on the physically meaningful selection of these twelve model parameters, their values were varied within a set of model simulations. The intention of these model simulations was to derive the interrelationship between model parameters and performance criteria. For this,** model simulations for the period from 2000 to 2010 were carried out based on 2000 different parameter sets that were generated with the Latin Hypercube sampling approach as it is implemented in the r-package FME (Soetaert and Petzoldt, 2010). In the Latin Hypercube sampling, all model parameters were changed simultaneously within the whole parameter space. For a more detailed description, readers

are referred to Pfannerstill et al. (2014b).

**All parameters values were already in a hydrologically plausible range according to prior modelling experience with the study sites. These constrained parameter ranges allowed selecting an efficient but appropriate number of simulations to perform our analyses. Please note, that the intention of the presented study was not to identify the parameter values exactly, which allowed us to keep the sampling of the parameter space relatively sparse. Instead, we aimed to test and suggest the new connective strength approach. For this purpose, the number of 2000 model runs ensured a sufficient number of combinations at each node of the RTs.**

Moreover, we added a sentences to the description of regression trees (P.8, L.26-27):

**In our case, the set of 2000 parameter sets and performance criteria are large enough to ensure a sufficient number of results at each node of RT.**

Second reviewer:

After reading the revision and the responses to comments of all three reviewers. I'm kind of disappointed for several reasons. Therefore, I would rather suggest reject from publication at this point.

1. Authors mentioned in the response that the "focus of the manuscript is to investigate the relationship between model parameters and performance measures. Thus, we show which model parameters impact which performance measure and which performance measures are influenced by the different model parameters". I agreed, that's what I thought too. In my definition, that's also a part of parameter uncertainty, sensitivity analysis, and model calibration. One cannot investigate parameter uncertainty, sensitivity analysis by not doing what you did right (maybe not use LHS but other methods)? And, in most cases, that's a part of the work for model calibration. Then, scientists can use the calibrated model for further purposes, such as ANSWERING SCIENTIFIC QUESTIONS. That's why, I mentioned previously that the novelty of this manuscript is questionable.

We understand your remark as a general remark to the overall classification of our manuscript.

Following up on our earlier response, we do not aim with our current manuscript to "work" with the calibrated models and answer scientific or applied questions in the respective modelled watersheds. Instead our work is meant to be a contribution in the ongoing struggle of improving the modelling procedure itself, in particular to achieve a better handling of model parameters towards precise parameter value identification. We see our presented method as an additional and helpful approach to obtain knowledge about the suitability of different performance criteria that may be best to identify appropriate values of model parameters. Thus, following on our approach, the identification of the parameter values themselves could be the next step.

It seems that you agree with us, that there are other methods around that are likewise contributing to those issues. Against this background, we contribute with our work technically by adding a new approach (RT-analysis of the LHS simulations) and conceptually by introducing 1) the concept of bij*ective identifiability of parameter and performance measures (connective strength) and subsequently 2) the differentiation of* global sensitivity analyses and the connective strength to the discussion (Thanks for this nice and brief summary to the third reviewer (Richard Arsenault)). Thus, to our knowledge, there are no other approaches published, that consider explicitly the bijective relationship of model parameters and performance measures as we did in our manuscript.

To clarify this, we added this aim at a couple of places to our manuscript (P.3, L.30-32; P.9, L.15-16; P. 13, L. 13-17; P.15, L. 13-16).

We want to emphasize ones more, that we ourselves do not see our work as the final solution of parameter identification in hydrological modeling. Moreover, we see our approach as an alternative to sensitivity analysis and not as replacement. Rather, we see our work as contribution in an ongoing discussion as we stated in the last paragraph of the introduction of our manuscript (P.4, L.5-6):

"We present a way to detect the appropriateness of performance criteria **that are most helpful in the** identification of **hydrologically sound** model parameter **values**."

We hope that we got your point and that we could clarify the overall classification of our manuscript and where we see specifically the contribution of our work to the discussion of hydrological model evaluation.

2. In this study, some parameters were identified to be more influential than others. If you change the targeted watersheds, you'll definitely have different sensitive parameter sets and values. It is for sure technically-oriented, and I don't see the solid scientific values here.

We agree that for (the modeling of) different watersheds, different parameters will be more influential than others due to different dominances of different hydrological processes. The same holds for the suitability of performance measures to assess the model performance for different watersheds and their characteristics and the respective relationships of performance measures and parameters, which we termed "connective strength" in our manuscript.

Thus, our work had not the intention to present relationships of parameters and performance measures that are generally valid independent of the watershed, instead we want to contribute to the task of identification of most influential parameters in modeling with a generally applicable framework, that allows identifying and comparing characteristic parameter-performance measure relationships of different watersheds. However, we like to mention that some statements such as the strong bijective relationship between soil and evaporation parameters to performance criteria related to water balance, e.g. KGE_beta or RSR for mid flows are expected to be reproducible in other studies.

3. By the way, I just don't understand why you just don't add a map for both watersheds? How difficult is that?

We added a map of both catchments.

4. For what you did, I also think that's a failure. The reason is simple, and I didn't catch that in the first round but the reviewer#1 did. It was mentioned by reviewer#1 that "It is surprising that no appropriate performance criteria is found to relate to CN2, which is generally a sensitive parameter in SWAT". You justified that "CN2 was varied in these HRUs from 40 to 60 which is assumed to be sufficient to maintain the landscape heterogeneity". It seems the range of CN2 has been allocated in values that's already been validated in your previous work. In this case, of course the value of CN2 is not sensitive (it's in close to calibrated values anyway). Then, why are we even doing the proposed work in the first place? Are we supposed to explore the potential ranges (and the relationships between performance measures and parameters) of the parameters to validate the following steps? How can we know if other parameters are not being handled the same way?

We are not quite sure, whether we get the point, but we did not aim with our study to provide a global sensitivity-analysis of the parameter set of the SWAT model itself, but to investigate the parameter-performance measure relationships of the hydrologically realistic model runs. Thus we did not randomly vary all parameters of the SWAT model. Instead we restricted our analysis to ranges of parameter values which are according to our knowledge of the study

areas hydrologically plausible. Therefore, the ranges of all parameters were carefully selected with the aim to represent the hydrological behaviour in a realistic way. E.g. CN2 was initially set to different values according to the land use type in the HRU, i.e. higher values for urban areas, lower for forest etc.. Based on these initial settings, CN2 was varied. Thus, certainly spatial heterogeneity is included and it does not make sense from hydrological point of view to increase or decrease CN2 dramatically. In addition it is not useful to increase CN2 above 100 or below 25 which since is beyond the possible values of CN2. Thus, if aiming to maintain spatial heterogeneity in landscape we have to set the parameter variation so that CN2 is within these ranges for all landscapes.

There are still several other issues, but I would like to stop it here. Hope you understand that I love your work, but the issues of novelty presented in this study cannot be resolved in easy ways.

Thank you for this final positive feedback. We interpret your statement like this that although you are not really convinced from our work at the current stage you still see some glance of novelty in it and that your main concern is the classification issue raised in your first remark. We hope that we could clarify this issue with our response to your first remark

Third reviewer (Richard Arsenault)

First, I commend the authors for their work on the revised manuscript. I believe the authors responded adequately to the reviewers comments and that the paper should be published in HESS. While I think that there might not be many uses of the proposed technique in the short-term, the idea of bijective identifiability is indeed novel and scientifically sound, and could open the door to further improvements. The differentiation between global sensitivity analyses and the connective strength approach is clearer. Finally, the limitations of the paper are better defined. Overall a good job by the authors.

Thank you very much for this very positive feedback and the very subtle and precise summary of our work!